# Automatic phenotyping using exhaustive projection pursuit

Wayne A. Moore [1,5] ✉, Stephen W. Meehan[1,5], Connor Meehan [2], David R. Parks [3], Guenther Walther [4] & Leonore A. Herzenberg [1]

One of the most common objectives in the analysis of flow cytometry data is the identification and delineation of phenotypes, distinct populations of cells with shared characteristics in the measurement dimensions. We have developed an automated tool to comprehensively identify these cell populations by Exhaustive Projection Pursuit (EPP). The method evaluates all two-dimensional projections among the suitable data dimensions and creates an optimized sequence of statistically significant gating regions that identify all phenotypes supported by the data. We evaluate the results of EPP on four well characterized data sets from the literature. The C++ code for EPP can be called from any computing environment. We illustrate this with a MATLAB utility that integrates EPP with FlowJo. All source code is freely available.

It is common in cell biology for cells to enter more-or-less stable states in which they express regulated amounts of specific proteins and other markers on their surface or internally. These can be labeled with reagents and their levels measured by flow cytometry. The resulting populations are identified with "flow phenotypes" based on representative expression levels of each marker. Therefore, a common analytic technique in both fluorescence and mass cytometry is phenotyping, identifying the phenotypes present in a sample and assigning individual cells to those types.

Historically this has mostly been done manually, using the methodology of "projection pursuit"[1]. Projected onto an appropriate plane, the data values associated with individual cell events may be grouped into different regions separated by empty space or a detectable valley so that a well-defined separation can be made. Ideally, these regions represent distinct populations or at least contain clusters with statistically meaningful separations. In current practice, a visual representation of the two-dimensional distribution is displayed for the user and "gates" are drawn manually around certain regions to identify phenotype markers. This process is repeated on each subset with a different pair of dimensions until no clear separation is possible in any pair of dimensions. Different choices in the analysis order should ideally all yield the same final phenotypes, albeit with some variability in assigning individual cells to phenotypes.

As more markers have been added, the number of pairs has grown rapidly. This has necessitated reliance on expert knowledge to choose appropriate pairs at each stage and to draw the gates. We present here an Exhaustive Projection Pursuit (EPP) method, which does not require prior knowledge and examines all pairs of dimensions to define a preferred gating at each stage.

Theoretically there are distributions that cannot be separated cleanly using only pairs of dimensions from a fixed set, and this is observed in raw fluorescence data. However, unmixed or compensated marker dimensions represent largely independent molecular features or distinct attributes of the cells that, in typical phenotyping situations, do not lead to complex population shapes that cannot be separated clearly in two dimensions. Historically, the vast majority of analysis of flow cytometry data has used two-dimensional projection pursuit methods, and this approach has been very successful in defining cell populations in flow cytometry. Furthermore, as long as clean two-dimensional splits are used, they are also correct for higher dimensional cases and correctly simplify the classification without impairing in any way the use of higher dimensional methods to further divide the identified populations.

Projection pursuit analysis has been standardized in Gating-ML[2] and is the basis for most software in the field. Modern cell sorters can implement sorting directly from such an analysis. These analyses are ubiquitous in the literature and widely understood. However, manual projection pursuit is time consuming and subjective. Criteria for selecting boundaries and the order of gating steps are ill defined and decisions may be influenced by expectations. Manual gating is not guaranteed to find the best separations, and it may not find all of them.

EPP avoids the limitations of manual projection pursuit analysis by automatically evaluating two-dimensional projections in all relevant pairs of dimensions in order to identify all statistically supportable phenotypes. In each set of two-dimensional projections, the algorithm locates event clusters and gating boundaries and selects the clearest available separation to divide the sample as the next step in the analysis. For both new subpopulations, the

[1]Department of Genetics, Stanford University School of Medicine, Stanford, CA, USA. [2]Independent Scholar, British Columbia, Canada. [3]Center for Molecular and Genetic Medicine, Stanford University School of Medicine, Stanford, CA, USA. [4]Department of Statistics, Stanford University, Stanford, CA, USA. [5]These authors contributed equally: Wayne A. Moore, Stephen W. Meehan. ✉e-mail: wmoore@stanford.edu

process is repeated until no statistically supportable separation is found, making each final population an algorithmic phenotype. EPP produces analyses that are otherwise entirely conventional and can be imported into existing software and sorted on or reviewed at will using familiar tools. However, it has the advantages of objectivity in selecting gates and gating paths and completeness in identifying all statistically supported phenotypes.

We have validated the EPP approach by comparing EPP results to 50 phenotypes identified in four published, peer reviewed studies using both fluorescence and mass cytometry. We selected three papers from the Cytometry Part A OMIP (Optimized Multicolor Immunofluorescence Panel) series, OMIP-044 by Mair, et al.[3], OMIP-047 by Liechti, et al.[4], and OMIP-077 by Boesch, et al.[5]. Each of these papers includes a well-developed reagent panel with a detailed recommended gating strategy for identifying the cell phenotypes of interest. Our fourth comparison reference is a mass cytometry study by Eshghi, et al.[6], using manual gating of a well-understood high-parameter panel for comparison to t-SNE[7]-guided manual gating.

## Results

### The EPP algorithm

Here we summarize the key elements needed to provide accurate and computationally efficient phenotyping by Exhaustive Projection Pursuit. A diagrammatic presentation is provided in Fig. 1. The EPP algorithm presumes that compensation or unmixing and selection of appropriate display transforms have been done. Preliminary gating for live single cells and dump channel exclusion is recommended, to avoid clutter with populations that are not relevant, like dead cells. Flow cytometry marker-reagent panels often provide some measurements for defining phenotypes and other measurements intended to characterize the properties of the events in defined phenotypes. EPP can identify phenotypes using just the former measurements with the other markers employed for subsequent analysis of the phenotypes.

The incremental nature of projection pursuit means that the errors are cumulative. If a split is made on a boundary where two biological phenotypes overlap in the measured dimensions, then each of the subpopulations will include events that are misallocated with respect to their biological type. These can only degrade the ability to further resolve the biology of the subpopulations. If subsequent steps also have overlaps, they will add additional misallocated events and never correct existing ones, so misallocations accumulate. Therefore, the analysis that chooses the smallest error at each step will have the fewest misallocated events at the end. In general, the true error is unknown but, other things being equal, it will scale as the number of events close to the boundary, so that can be used as a proxy.

At each stage in EPP analysis, data in some dimensions may be relatively uninformative and evaluation of two-dimensional distributions can be made more efficient by omitting them. In fluorescence cytometry, single positive or negative population distributions in "loggish" transforms like logicle[8] tend to be near normal. For mass cytometry with arcsinh transforms, positive distributions tend to be near normal, but negative population distributions are dominated by the way mass cytometry data is processed. For low signal measurements, the data values include numerical zeroes and small integers. In our one mass cytometry data set, distributions of non-zero values in negative dimensions are roughly exponential.

EPP first tests individual dimensions to qualify them for use by looking for deviation from normal or exponential using Kullback-Leibler Divergence (KLD)[9] and omits those with insufficient divergence. For each pair of remaining qualified dimensions, a Gaussian kernel density estimate is prepared, to which 2-D Modal Clustering is applied. This produces the cluster graph, a plane graph composed of vertices (points) where three or more edges meet, edges that each connect two vertices and must not intersect, and faces bounded by three or more edges. Each face contains one isolated mode or local maximum of the density. Density-Based Merging is done by finding the density at the highest point along each edge and testing it for a statistically significant difference from the estimated density at the modes of the clusters the edge separates[10]. Those without a significant dip are removed, leaving the separable clusters.

When there are multiple clusters remaining, EPP always divides the sample into two parts and omits the other possible separations. The data in the subpopulations are unchanged and, at a later stage, the algorithm will rediscover them and have another chance to select the same edges. To achieve this efficiently, Graph Simplification using dual graph methods[11] is used to find continuous edges separating two contiguous regions, suitable for use as gates, which are scored by estimating the number of events near the boundary.

When all suitable 2 Part DBM and Simplified graphs have been processed, in "best separation" mode the separation with the fewest events close to the boundary, i.e., the one as far removed from the dense clusters as possible, is selected to divide the population. Alternatively, in the "best balance" mode used here and described in the Methods section, the selection is weighted to favor more equal splits. Best separation tends to initially pick out small populations while best balance is more similar to typical human selections in identifying larger populations early but may miss very small small ones if they lose significance.

The sample is then Divided by The Best Scoring graph and EPP is applied recursively to each branch until no additional splits are found. At that point, the resulting final populations are designated as EPP "leaves" or algorithmic phenotypes.

All these computations proceed in parallel in the current version of EPP which is implemented in C++. The resulting complete analysis is available as a JSON[12] encoded conventional "gating tree". The MATLAB bridge can be used to run EPP on a chosen subpopulation from a FlowJo workspace and then augment the workspace by adding the EPP gating tree. The EPP populations and gates can be displayed and used as in any other FlowJo gating tree (e.g., see Fig. 2). EPP gating sequences can be implemented on standard instruments for cell sorting. The JSON also includes the score for each gating tree boundary. A separate CSV file contains the assigned numerical cluster identifier for each observation as well as the Mahalanobis distance of the observation from the cluster center.

Computers currently used for FlowJo, R or MATLAB are suitable to run EPP. The implementation we used is not fully optimized, but nevertheless runs in seconds to minutes on the data sets used, so performance will not be a barrier. It is designed to take full advantage of modern multi-threaded CPUs and the design anticipates distribution over clusters as well. The source code is available on GitHub[8] under the BSD license.

### Finding candidate separations

When there are more than two separable clusters, we need to identify suitable candidate separation boundaries from some but not all of the edges in this graph. Each candidate should be a continuous curve, composed of edges of the cluster graph and separating two contiguous regions. We have developed what we think is a previously unknown algorithm to enumerate these candidate boundaries efficiently. The graph-theoretic dual of the cluster graph exchanges vertices for faces, while edges map to edges. We can remove one edge of the dual graph and merge its two vertices, so that every edge ending in either of the vertices now ends in the new merged vertex. The corresponding operation for the cluster graph to preserve duality is removing an edge and combining two faces, i.e., merging contiguous clusters. Consistency will sometimes require that we also merge other dual edges because they now connect the same two vertices. The corresponding operation for the cluster graph is splicing together two cluster edges at a common vertex, replacing them with a single new edge without that vertex. This step always reduces the faces by one and the edges by one or more and in the latter case also the vertices, thus simplifying the graph. This elementary step can be compounded systematically to find all simplified dual graphs with only one edge, each of which is either the dual of a single cluster edge or was constructed as above by merging the duals of the cluster edges and is always dual to a candidate boundary. These operations on the dual graph are entirely abstract, so they can be implemented very efficiently.

### Comparison with published phenotyping experiments

EPP analysis is started after the published pregating and subsequently uses the same markers as the publication. In an EPP gating sequence the

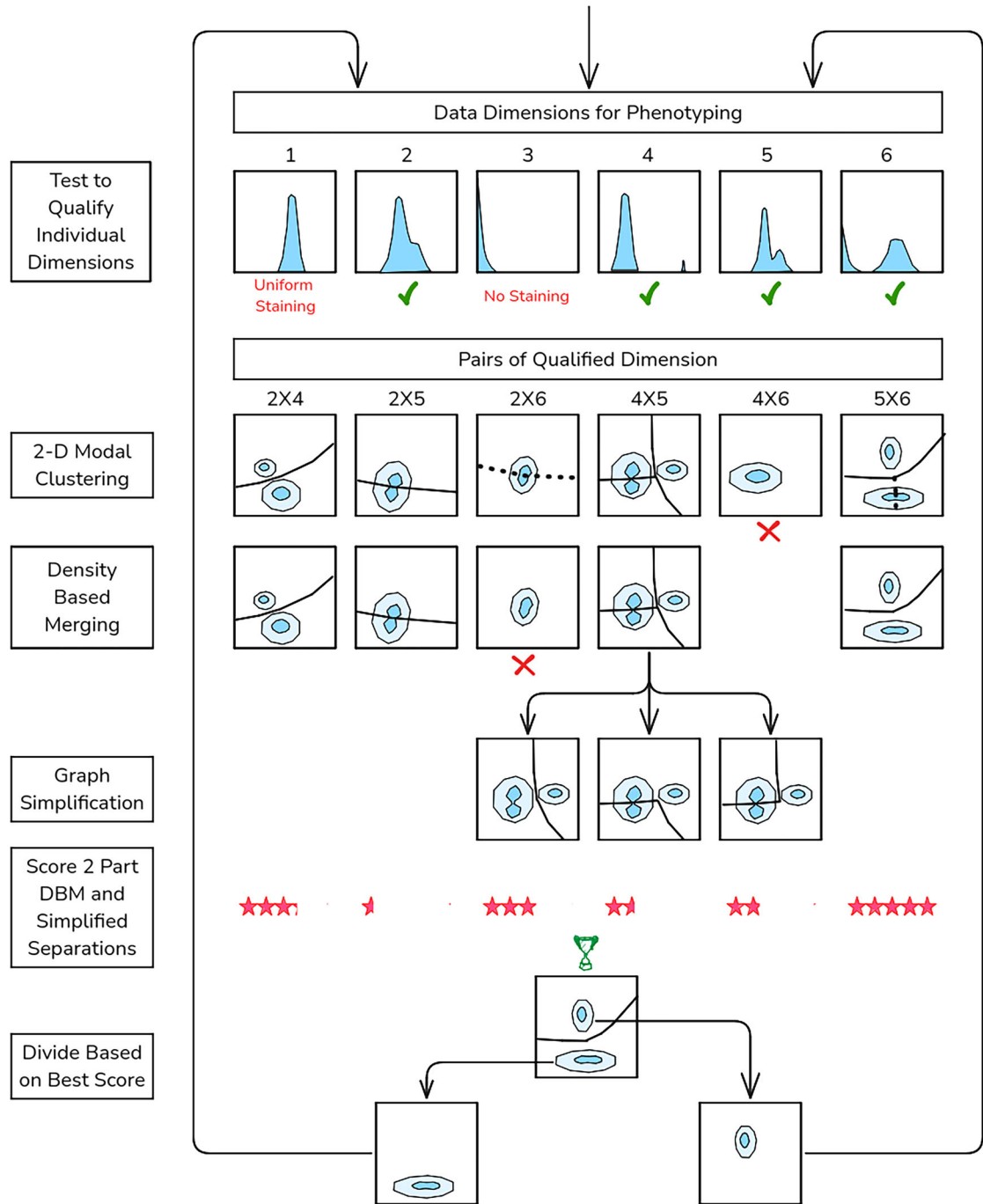

**Fig. 1 | EPP Algorithm.** Diagram of the EPP gating analysis procedure. Data Dimensions are tested to see if they Qualify as likely to be useful in distinguishing different cell populations. Qualified Pairs are enumerated and 2-D Modal Clustering is applied to each pair to delineate candidate separation boundaries. Candidate separations are tested with Density-Based Merging, and statistically valid ones are retained. Graph Simplification is applied where there are more than two clusters. The accepted two-part separations are scored, the Best Scoring among all the reduced graphs is selected, and its two parts are each returned to the Qualify Individual Dimensions stage. In each branch, the process ends when no valid separations are found after Density-Based Merging.

algorithmic phenotypes, the final populations sometimes designated as "leaves", should be final phenotypes in the sense that they cannot be further divided into statistically supportable subpopulations using the data. Final populations in conventional manual gating may or may not be final phenotypes as defined here. The EPP labels are constructed heuristically based on the FCS keywords and average signals in order to make them somewhat comprehensible and should not be taken as a biological assessment.

To compare the EPP results to the populations defined in the reference analyses, they are first aligned using QFMatch[13], and the overlapping events between published populations and EPP algorithmic phenotypes are enumerated. We identify and evaluate cases where EPP splits published final populations into several phenotypes and cases where EPP does not support a population split defined in the reference gating. In addition, there are cases where EPP identifies phenotypes omitted in the reference gating. A Match Table showing counts of matching events between the EPP and reference populations was constructed for each of the data sets.

**OMIP-077—Boesch, et al**. For OMIP-077[5], Boesch, et al., designed a 14-color panel to define "all principal human leukocyte populations" in whole blood samples. Preliminary gating included CD45+, FSC-A x

**Article**

SSC-A cells, FSC-A x FSC-H single cells and SSC-A x CD34 for Leukocytes. Our recreation of the reference gating scheme is shown in Supplementary Fig. 1. The gating scheme for the EPP analysis is shown in Fig. 2. A table comparing the EPP and Reference analysis results is in Fig. 3. Marker-Dye combinations are documented in Supplementary Fig. 2.

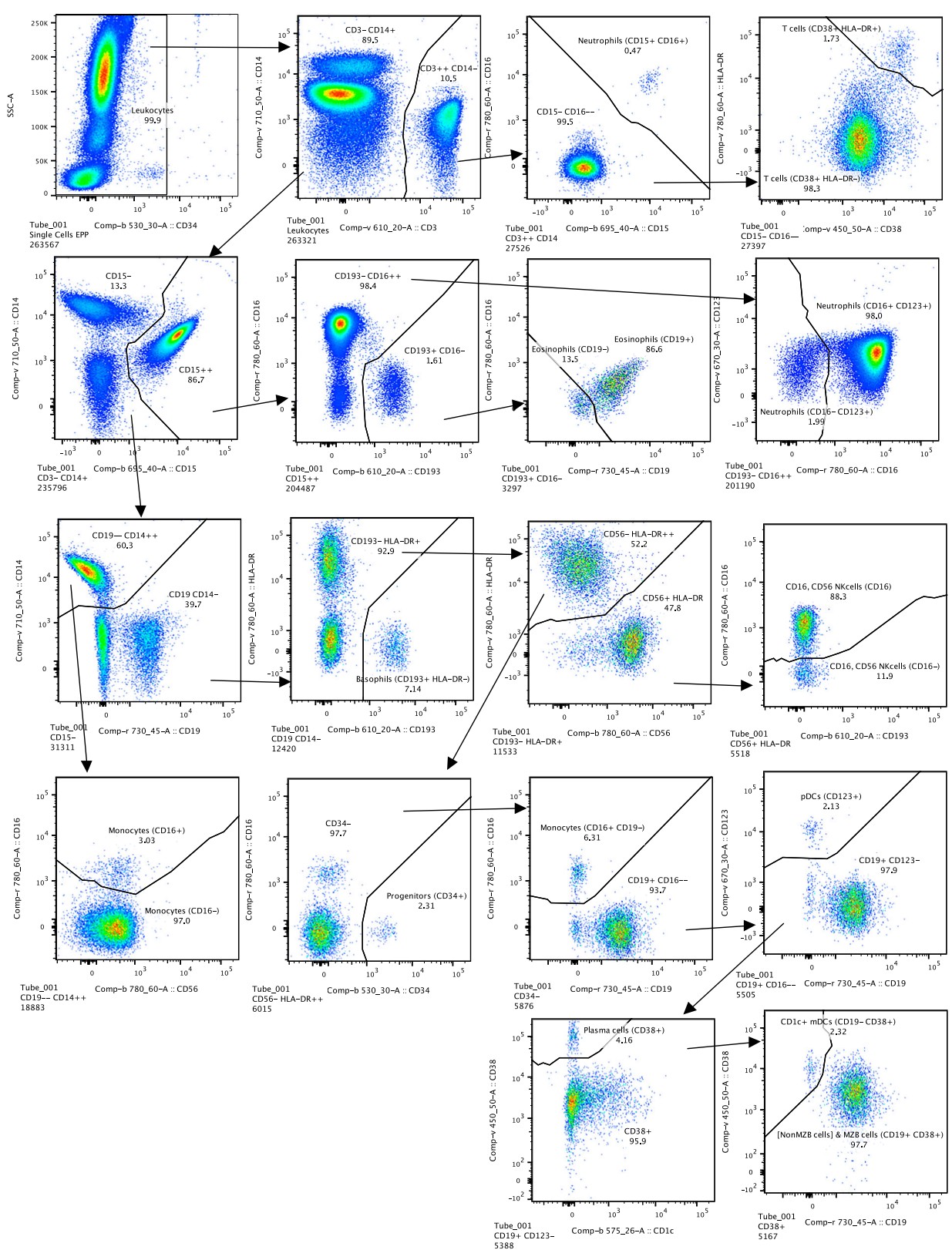

**Fig. 2 | OMIP-077 EPP Gating.** OMIP-077 EPP gating sequence starting from the reference "Leukocytes" population. The recreation of the OMIP-077 reference gating in Supplementary Fig. 1 shows the Leukocytes population in the Row 1 Column 4 panel. The marker-dye-primary detector alignments for the OMIP-077 reagents are listed in Supplementary Fig. 2.

| EPP populations | events | 1 Neutrophils 201,691 | 2 T cells 27,847 | 3 Monocytes 18,228 | 4 NKcells 5,149 | 5 [Non NK (CD16-CD56-)] 282 | 6 [Non MZB cells] 3,723 | 7 MZB cells 1,362 | 8 Eosinophils 3,329 | 9 Basophils 889 | 10 Plasma cells 227 | 11 Progenitors 181 | 12 pDCs 112 | 13 CD1c+ mDCs 74 | 14 [CD141- CD123- mDCs] 112 | 15 CD141+ mDCs 27 | Not matched |
|---|---|---|---|---|---|---|---|---|---|---|---|---|---|---|---|---|---|
| 1 Neutrophils(CD16+ CD123+) | 197,175 | 197,079 | | 28 | 3 | | | | 8 | | | | | | | | 57 |
| 2 Neutrophils(CD16- CD123+) | 3,977 | 3,915 | | 13 | 12 | 10 | 15 | 3 | | | | | | | 2 | | 7 |
| 3 Neutrophils(CD15+ CD16+) | 131 | 128 | | | | | | | | | | | | | | | 3 |
| 4 T cells(CD38+ HLA-DR-) | 27,427 | 6 | 27,371 | | | 6 | | | 2 | | | | | | 1 | | 41 |
| 5 T cells(CD38+ HLA-DR+) | 494 | | 475 | | | | | | | | | | | | | | 19 |
| 6 Monocytes(CD16-) | 18,311 | 421 | | 17,319 | | | | | 7 | 5 | | 17 | | 7 | 12 | 6 | 517 |
| 7 Monocytes(CD16+) | 557 | 15 | | 501 | | | | | 1 | | | 2 | | | | | 38 |
| 8 Monocytes(CD16+ CD19-) | 378 | | | 335 | 14 | | | | | | 1 | 1 | | | 11 | 2 | 14 |
| 9 NKcells(CD16+) | 4,928 | 86 | | 12 | 4,787 | | | 1 | | | | | | | 3 | | 39 |
| 10 NKcells(CD16-) | 672 | 25 | | | 332 | 249 | 3 | 1 | | | | 24 | | | 16 | | 22 |
| 11 [NonMZB cells] & MZB cells(CD19+ CD38+) | 5,182 | | | 11 | | 6 | 3,680 | 1,347 | | 3 | | 11 | | 12 | 20 | | 92 |
| 12 Eosinophils(CD16- CD193-) | 3,301 | 6 | | | | | | | 3,288 | 2 | | | | | | | 5 |
| 13 Basophils(CD193+ HLA-DR-) | 909 | | | | 1 | 6 | | | 23 | 874 | | | | | | | 5 |
| 14 Plasma cells(CD38+) | 219 | | | | | 3 | | | | | 213 | | | | 1 | | 2 |
| 15 Progenitors(CD34+) | 139 | 1 | | | | | | | | | | 137 | | | | | 1 |
| 16 pDCs(CD123+) | 115 | | | | | | 2 | 1 | | | | | 112 | | | | |
| 17 CD1c+ mDCs(CD19- CD38+) | 125 | | | | | 1 | | | | | | | | 55 | 46 | 19 | 4 |
| Not matched | | 9 | 1 | 9 | | | 23 | 10 | | 5 | 2 | | | | | | |

**Fig. 3 | OMIP-077 Match Table.** Table of events matches between OMIP-077 Reference populations and EPP phenotypes. The Column headers show the manually gated Reference populations and their event numbers. The Rows show the EPP-identified populations with cell type names from their matched Reference populations and indicators of markers used in the EPP separation. The associated +/- levels are based on population positions in the EPP graphs. The main table cells show the number of shared events between the corresponding Reference and EPP populations. The GREEN highlights indicate dominant matches. The ORANGE cell indicates the smaller component of an EPP population that was divided between two Reference populations.

The reference gating illustrated in Supplementary Fig. 1 delineated 12 cell populations. Three additional populations were clearly visible in the sequence but not specifically noted, so we added gates for them ([Non MZB cells], [Non NK(CD16- CD56-)] and [CD141- CD123- mDCs]). The EPP analysis led to 17 leaf populations. One low frequency population designated Neutrophils(CD15 + CD16+) likely consists of Neutrophil-T cell doublets. The reference gating indicates three very low frequency populations of mDCs (CD1c+ mDCs, [CD141- CD123- mDCs] and CD141+ mDCs) that seem to be based on expectation rather than any indication of separable populations in the distribution of the data. EPP identified just one well-defined mDC population CD1c + mDCs(CD19-CD38+) including CD141+ and CD1c+ events but detected no justifiable splits among the mDCs. The reference gating designates marginal zone B (MZB) cells as CD1c+ with no sign of separation between these and the main B cell population we designated as [Non MZB cells]. EPP identifies essentially the same overall set of B cells but did not have a basis for defining a CD1c+ MZB phenotype, so both MZB cells and [Non MZB cells] appear in a single EPP leaf [NonMZB cells] & MZB cells(CD19 + CD38+) (Fig. 3, Row label 11) which almost exactly matches the reference MZB + [Non MZB cells]. We note that the opt-SNE plot in the OMIP-077 Figure 1[5] does not show a separate MZB cell population.

The other 14 EPP leaves each matched one of the other 10 reference populations with good correspondence between the event counts of the reference populations and the sum of their associated EPP leaves. Among the reference T cells, EPP identified the main T cells(CD38 + HLA-DR-) leaf and also a small T cells(CD38 + HLA-DR+) leaf. The reference gating does not show this population, but the authors state that the stain panel "allows for the quantification of HLA-DR and CD38 double-positive cells".

The OMIP-077 panel was designed to provide clear discrimination of the major cell populations, and EPP did well in delineating them and, in some cases, defining further subpopulations.

**OMIP-044 - Mair, et al**. In the OMIP-044[3] paper, Mair, et al., created a 28-color panel designed for analysis of antigen presenting cells in peripheral blood mononuclear cell (PBMC) samples but including markers for identifying the other major phenotypes. Preliminary gating used FSC-A × SSC-A, FSC-A × FSC-H and CD45 × live-dead to select for live, single CD45+ cells. The upper three rows in Supplementary Fig. 3 show our recreation of the main gating structure illustrated in Figure 1[3] of Mair, et al., that defines the ten basic phenotypes of interest. We added three gates ([not naive CD4+], [not naive CD8+] and [not DCs]) to capture regions that were apparent but not labeled in the reference gating structure. The EPP gating is shown in Supplementary Figs. 4 and 5, and the matching table is in Fig. 4. Marker-Dye combinations are documented in Supplementary Fig. 2.

The EPP analysis led to 40 distinct leaves. Six low-frequency EPP leaves (Fig. 4, Rows 34-39) seem to be cell doublets that were not excluded in the single cells gating. In the reference gating, three of these are mixed into other defined populations while the other three are unclassified.

Among the remaining 34 EPP leaves, 23 align predominantly into one of the ten original reference populations, 9 align to one of the added [not] gates, one falls outside any of the reference populations, and one is about evenly split between the Naive CD4 and [not Naive CD4] reference populations. In this last case, the low signal cutoff on CD45RA in the Naive CD4 gate shows no separation, so it is not surprising that the EPP leaves [not naive CD4+](CD45RA- CCR7+) includes events above and below that cutoff and contributes about equally to Naive CD4 and [not Naive CD4]. The original reference gating defines a CD123+CD56- population that includes both plasmacytoid DCs (pDCs) and basophils. In a corrigendum to the OMIP-044 paper[14], Mair, et al., recommend a revised gating to distinguish plasmacytoid DCs (pDCs) as CD56-HLA-DR+CD11c-CD123+ from CD56-HLA-DR-basophils as illustrated in the fourth row of Supplementary Fig. 3. EPP uses a different gating strategy including CD45RA to identify two quite distinct HLA-DR+CD45RA+ and HLA-DR-CD45RA- leaves that correspond to the intended pDCs and basophils, respectively.

Overall, EPP identifies one to six phenotypes in each of the ten primary reference populations that together represent the large majority of the events.

**OMIP-047—Liechti, et al.** For OMIP-047[4], Liechti, et al., developed a 16-color panel for "detailed dissection of human B cell subsets and their phenotype in peripheral blood mononuclear cells (PBMC)". Preliminary gating included a time gate, FSC × SSC for cells, FSC-A × FSC-H single cells, dump exclusion of non-B cell lineages and CD19+ gating to select the B cell population. In addition to markers defining common B cell

| EPP populations | events | 1 [not naive CD4+] 339,632 | 2 naive CD4+ 453,746 | 3 CD14+ 371,366 | 4 CD19+ 252,670 | 5 CD56+ NK 144,124 | 6 [not naive CD8+] 105,744 | 7 naive CD8+ 131,064 | 8 DN DCs 51,976 | 9 DN 53,390 | 10 CD123+ pDC 23,462 | 11 CD1c DCs 8,222 | 12 [not DCs] 16,476 | 13 CD141+ DCs 908 | Not matched |
|---|---|---|---|---|---|---|---|---|---|---|---|---|---|---|---|
| 1 [not naive CD4+](CD56-) | 222,407 | 213,971 | 2,866 | 275 | | 1 | 3 | | | 2,026 | 8 | | 1,999 | | 1,258 |
| 2 [not naive CD4+](CD45RA- CCR7+) | 110,747 | 65,088 | 43,276 | 239 | | | | | | 1,481 | | | 228 | | 435 |
| 3 [not naive CD4+](CCR7-) | 62,239 | 58,968 | 19 | 152 | | 21 | 1 | | | 280 | 14 | | 1,656 | | 1,128 |
| 4 [not naive CD4+](CCR7-) | 340 | 317 | | 2 | | | | | | 1 | | | 10 | | 10 |
| 5 [not naive CD4+](CD56+) | 251 | 223 | 1 | 1 | | 2 | | | | 4 | | | | | 20 |
| 6 naive CD4+(CCR7+) | 412,566 | 990 | 406,961 | 712 | | 1 | | 6 | | 1,978 | 8 | | 705 | | 1,205 |
| 7 CD14+(CD1c- CD14+) | 341,753 | | 1 | 340,412 | 4 | 396 | | | 614 | | 1 | 18 | 3 | | 304 |
| 8 CD14+(CD45RA- CD14+) | 24,015 | | | 22,829 | 3 | 1 | | | 895 | | | 1 | | | 286 |
| 9 CD14+(CD14+ CCR7-) | 142 | | | 142 | | | | | | | | | | | |
| 10 CD19+(CD11c- CD56-) | 252,905 | | | 875 | 251,607 | | | | 1 | | | | 28 | | 394 |
| 11 CD56+ NK(CD11c+) | 95,842 | 2 | | 96 | 5 | 94,480 | | 2 | 44 | | | | 1,025 | | 188 |
| 12 CD56+ NK(CD11c-) | 36,018 | | | 1 | 1 | 35,430 | | | | | | | 555 | | 31 |
| 13 CD56+ NK(CD56+) | 7,328 | 2 | | 95 | 3 | 6,437 | | | | 1 | | | | | 790 |
| 14 CD56+ NK(CD16- CD56+) | 3,091 | | 1 | 1 | | 3,078 | | | | | | | | | 11 |
| 15 CD56+ NK(CD11c+ CD56+) | 865 | | | 170 | 3 | 660 | | | | | | | 15 | | 17 |
| 16 CD56+ NK(HLA-DRlo CD141-) | 265 | | | 1 | | 206 | | | 48 | | | 2 | 6 | | 2 |
| 17 [not naive CD8+](CCR7-) | 57,574 | | | 131 | | 81 | 53,164 | 6 | 1 | 105 | | | 2,241 | | 1,845 |
| 18 [not naive CD8+](CD4-) | 40,637 | | | 62 | | 1 | 37,993 | 1,636 | | 37 | | | 565 | | 343 |
| 19 [not naive CD8+](CD56+) | 12,568 | | | 31 | | 349 | 11,893 | 1 | | 62 | | | | | 232 |
| 20 naive CD8+(CD8+) | 136,547 | | | 259 | | | 1,680 | 129,261 | | 546 | 1 | | 3,224 | | 1,576 |
| 21 DN DCs(CD45RA+ CD14-) | 54,020 | 2 | 6 | 3,775 | 12 | 58 | 2 | 1 | 47,991 | 10 | 134 | 12 | 32 | | 1,985 |
| 22 DN DCs(CD11c+ CD56-) | 1,153 | | | 5 | | 4 | | | 757 | | 2 | | 365 | | 20 |
| 23 DN(CD56+) | 20,719 | | | 25 | | 115 | 410 | 2 | | 20,131 | | | 1 | | 35 |
| 24 DN(CD3+ CCR7-) | 19,964 | | | 55 | 1 | 2 | 156 | | | 19,323 | 24 | | 323 | | 80 |
| 25 DN(CD8-) | 5,178 | | | 10 | | 2 | | | | 4,664 | | | 411 | | 91 |
| 26 DN(CD3+ CCR7+) | 2,092 | | | 12 | | 1 | 17 | | | 1,625 | 2 | | 365 | | 70 |
| 27 DN(CD45RA+ CD3+) | 1,283 | | | | | | 30 | 1 | | 1,056 | | | 162 | | 34 |
| 28 CD123+ pDC(HLA-DR- CD45RA-) | 16,560 | | | 66 | | | | | 1 | | 16,227 | | 132 | | 134 |
| 29 CD123+ pDC(HLA-DR+ CD45RA+) | 7,360 | | | 12 | | | | | 1 | | 7,013 | | 33 | | 301 |
| 30 CD1c DCs(CD45RA- CD56-) | 12,961 | 3 | 1 | 470 | 12 | 2,040 | | | 1,598 | 1 | 28 | 8,189 | 6 | 55 | 558 |
| 31 [not DCs](CD16- CD56-) | 1,684 | | | 6 | | 4 | 1 | | 24 | | | | 1,645 | | 4 |
| 32 [not DCs](CD11c- CD14-) | 784 | | | | 26 | 67 | | | | | | | 655 | | 36 |
| 33 CD141+DCs(CD141+) | 1,790 | | | 3 | | 640 | | | | | 1 | | 72 | 853 | 221 |
| 34 naive CD4+ doublets(CD45RA+ CD56+) | 830 | 63 | 609 | 21 | | 27 | | | | 34 | | | | | 76 |
| 35 CD19+ doublets(CD11c+ CD56+) | 511 | | | 2 | 500 | 1 | | | | | | | | | 8 |
| 36 naive CD8+ doublets(CD16+ CD56+) | 202 | | | 1 | | 18 | 10 | 140 | | 22 | | | | | 11 |
| 37 No Ref match doublets(CD123- CD3+) | 1,974 | | 2 | 22 | 4 | | | | | | | | | | 1,946 |
| 38 No Ref match doublets(CD4+) | 710 | | | 4 | | | | | | | | | | | 706 |
| 39 No Ref match doublets(CD8+ CD3+) | 531 | | | 7 | 24 | | | | | | | | | | 500 |
| 40 No Ref match(CD4) | 1,626 | | | 2 | | | 384 | 9 | | | | | 14 | | 1,217 |
| Not matched | | 3 | 3 | 383 | 464 | 1 | | | | 2 | | | | | |

**Fig. 4 | OMIP-044 Match Table.** OMIP-044 Match Table between reference populations and EPP phenotypes. The Column headers show the manually gated Reference populations and their event numbers. The Rows show the EPP-identified populations with cell type names from their matched Reference populations and indicators of markers used in the EPP separation. The associated +/- levels are based on population positions in the EPP graphs. The main table cells show the number of shared events between the corresponding Reference and EPP populations. GREEN highlights indicate dominant matches. The ORANGE cell indicates a second Reference population that contains part of an EPP phenotype. The two BLUE cells highlight the EPP phenotypes that split the reference CD123 + pDC population.

subsets, the reagent panel included additional phenotypic and functional markers. Our recreation of the reference gating in Fig. 1[4] of OMIP-047 is shown in Supplementary Fig. 6. The EPP gating sequence is illustrated in Supplementary Fig. 7. Figure 5 shows event matching between each of the reference populations and the EPP final populations.

EPP identified well-defined IgG1+, IgG3+, CD38+CD27++ and IgA populations dividing the IgG3+ population into CD27+ and CD27- populations that were not distinguished in the reference gating. The CD38+CD27++ population was not identified in the reference gating, but even though it included few events, there was sufficient separation for EPP to divide it into an IgD- leaf that constituted two thirds of the reference PB (plasmablast) population and an IgD+ leaf that had "No reference match". The reference PB gating showed no clear basis for a separation in either its IgD or CD38 dimension, and the other half of its events were distributed among four EPP leaves. After removing the

populations noted above, EPP identified a CD27+ leaf corresponding to the reference MZB cell population and a CD27- leaf including mostly Naïve B cells. The IgD distribution of the CD27- non-(IgA or IgG3 or IgG1) population did not show a usable divide between IgD+ and IgD-, so the EPP Naïve B leaf includes the reference Naïve B cells and the IgD- Non-IgA/IgG1/IgG3 Memory B population.

The reference gating on the B cell population started with selection of CD10+ B cells leading to an IgD+CD38+ Transitional population (see Supplementary Fig. 6). In EPP, CD10 and CD38 together or in combination with other markers did not provide an acceptable separation of populations, so the Transitional events remained among the MZB cell population. Subsequent gating at the high edge of an FMO distribution could provide a minimum estimate of positive events while excluding a portion of the ideal events, but the available data do not provide a way to accurately identify or enumerate a Transitional phenotype.

| EPP populations | events | 1 Naive 22,933 | 2 MZ B cells 16,518 | 3 IgA_IgG1_Ig 4,332 | 4 IgA+ 2,746 | 5 IgG1+ 1,788 | 6 Transitional 1,069 | 7 IgG3+ 377 | 8 PB 215 | Not matched |
|---|---|---|---|---|---|---|---|---|---|---|
| Transitional(CD27-) | 26,173 | 22,720 | 212 | 886 | 40 | 7 | 1,059 | | 29 | 1,220 |
| 2 MZ B cells + Non-IgA_IgG1_IgG3(CD27+) | 21,485 | 155 | 16,171 | 3,446 | 202 | 17 | 8 | | 15 | 1,471 |
| 3 IgA+(IgA+ IgD-) | 2,773 | 4 | 14 | | 2,504 | 1 | | | 6 | 244 |
| 4 IgG1+(IgG1+ IgD-) | 2,167 | 9 | 81 | | | 1,763 | 1 | | 24 | 289 |
| 5 IgG3+(CD27-) | 221 | 1 | 2 | | | | | 196 | 1 | 21 |
| 6 IgG3+(CD27+) | 208 | | 3 | | | | | 181 | 2 | 22 |
| 7 PB(IgD-) | 173 | | | | | | | | 138 | 35 |
| 8 No reference | 213 | | | | | | | | | 213 |
| Not matched | | 44 | 35 | | | | 1 | | | |

**Fig. 5 | OMIP-047 Match Table.** OMIP-047 Match Table between reference populations and EPP phenotypes. The Column headers show the manually gated Reference populations and their event numbers. The Rows show the EPP-identified populations with cell type names from their matched Reference populations and indicators of markers used in the EPP separation. The associated +/- levels are based on population positions in the EPP graphs. The main table cells show the number of shared events between the corresponding Reference and EPP populations. GREEN highlights indicate dominant matches. The ORANGE cells indicate a second Reference population that contains part of an EPP phenotype.

**Eshghi, et al.** In this paper[6] Eshghi, et al., compared conventional gating analysis to t-SNE-guided analysis using a 38-marker mass cytometry panel applied to human PBMCs. Five additional mass reagents were used for preliminary selection of live single cells. The reference gating scheme defined 28 named populations and included two gate regions that were unnamed for a total of 30 defined populations (see Fig. 1[6] in Eshghi, et al. or our recreation in Supplementary Fig. 8 where the 28 populations are numbered). The EPP analysis is illustrated in four parts as Supplementary Figs. 9, 10, 11 and 12. Figure 6 shows the shared events between selected major reference populations and the major EPP phenotypes. The full version showing all defined populations is in Supplementary Fig. 13.

EPP initially identified a set of 105 valid populations. Fourteen EPP phenotypes with event counts below 2000 were distinct from any of the reference populations. Three of the reference populations were found only as minority partners in an EPP phenotype, but all of the other reference populations had one to eight EPP phenotypes as primary associations. Of the 77 EPP phenotypes not associated with the poorly resolved CD4 EM and CD4 CM reference populations or with no reference match, all but three had dominant matching with only one reference population.

Notable observations on the relationships of reference populations and EPP leaves:

1. The single EPP "HLADRloMono+CD14hiMono(CD11bCD14+)" leaf (Fig. 6, Row label 1) includes a large majority of the events in the Ref-24 CD14hi Mono and Ref-27 HLADRlo Mono populations, and nothing in the EPP analysis distinguishes these reference populations from each other. When overlaid in an HLA-DR x CD11c display, these reference populations look like a single entity with no indication of an appropriate break point between lower and higher HLA-DR regions. These reference populations should be considered parts of the same population, and EPP puts them in the same leaf.

2. The Ref-11 CD4 CM and Ref-12 CD4 EM populations are represented in the EPP analysis by 9 significant leaves (Fig. 6, Row labels 17-23,37,38). Four of these substantially overlap both CD4 EM and CD4 CM, 7 are predominantly in CD4 EM, and two are predominantly in CD4 CM. In the reference gating (as shown in Fig. 1[6] in Eshghi, et al., or our recreation in Supplementary Fig. 8, Row 4 Column 5), the gate boundary is in an area of substantial event density with only a small dip

in the CCR7 distribution between CD4 CM and CD4 EM, so it is not surprising that five of the associated EPP leaves which were derived using better separations in other dimensions do not divide in the same way.

3. The EPP CD8 EM + CD8 CM(Va7.2-) leaf (Fig. 6, Row label 42) aligns mostly with Ref-7 CD8 EM (Fig. 6, Column label 9) but also includes most of Ref-6 CD8 CM (Fig. 6, Column label 26). In the reference gating, these populations are separated as CCR7- and CCR7+ without any sign of a break in the data (Supplementary Fig. 8, Row 4 Column 4), so it is not surprising that EPP does not distinguish them into separate leaves.

4. The event pattern in the Ref-13 CD4 Temra gate (Supplementary Fig. 8, Row 4 Column 5) shows no indication of a distinct population or separation from the adjacent Ref-10 CD4 N and Ref 12- CD4 EM populations. CD4 Temra (Fig. 6, Column label 21) seems to be an aggregate of overlaps since only one EPP phenotype (CD4 Temra(CCR7-), Fig. 6, Row label 79) with 7% of its events aligns with it and the rest of the Ref-13 CD4 Temra events are distributed among the 11 EPP phenotypes that align primarily with the Ref-10 CD4 N, Ref-11 CD4 CM and Ref-12 CD4 EM populations. The projection of the manually gated CD4 Temra population onto the t-SNE space in Fig. 3B[6] of Eshghi, et al. supports this interpretation. It shows CD4 Temra events scattered among other cell types in the t-SNE space indicating considerable diversity within the CD4 Temra assignment.

**Summary comparisons between EPP and reference populations.** Figure 7 compares the event sets for the populations identified in the four reference studies with the EPP final phenotypes whose event sets are mostly assignable to particular reference phenotypes. We use two versions of the Jaccard Similarity as metrics for the match between reference and EPP populations. For the ordinary Jaccard Similarity coefficient, given two sets A and B, we compute the fraction of events in either set that are in both sets, i.e., $J(A, B) = \text{card}(A \text{ and } B)/\text{card}(A \text{ or } B)$. For the Central Similarity coefficient, we restrict the comparison to the central 80% of events in $A$ or $B$ based on the Mahalanobis distance of each point from the mean. A high value for the Central Similarity coefficient gives assurance that the different methods are identifying the same biological

| Reference populations → / EPP populations ↓ | events | 1 CD14hi Mono | 2 HLADRlo Mono | 3 CD4 N | 4 CD16hi NK | 5 CD4 EM | 6 CD4 CM | 7 CD8 Temra | 8 CD8 N | 9 CD8 EM | 10 Naive B | 11 Basophils | 12 CD16hi Mono | 13 CD8 MAIT | 14 gd Tcells | 15 Neutrophil | 16 int Mono | 17 CD56hi NK | 18 Mem B | 19 Trans B | 20 pDCs | 21 CD4 Temra | 22 Eosinophil | 25 mDCs | 26 CD8 CM | 27 Plasmablast | Not matched |
|---|---|---|---|---|---|---|---|---|---|---|---|---|---|---|---|---|---|---|---|---|---|---|---|---|---|---|---|
| (events) | | 80,750 | 11,310 | 70,143 | 56,449 | 45,354 | 29,702 | 29,765 | 23,383 | 16,756 | 13,016 | 11,906 | 11,776 | 8,886 | 9,049 | 7,929 | 8,126 | 3,298 | 3,041 | 3,179 | 2,519 | 8,120 | 2,409 | 1,833 | 2,266 | 1,350 | |
| 1 HLADRlo Mono + CD14hi Mono(CD11b CD14+) | 94,536 | 79,523 | 10,810 | | 3 | | | | | | 3 | 19 | | | | | 714 | | 10 | | | 3 | 95 | 11 | | 1 | 2,988 |
| 3 CD4 N(CD38-) | 58,567 | | | 55,148 | | | | | | | | | | | | 59 | | | | | | 1,819 | 16 | | | | 1,193 |
| 4 CD4 N(CCR7-) | 11,012 | | | 9,148 | | | | | | | | | | | | 18 | 1 | | | | | 1,395 | 2 | | | | 291 |
| 11 CD16hi NK(CD161-) | 18,267 | | | | 17,430 | | | | | 1 | | | | | | | 139 | 1 | | | | | | | | | 685 |
| 12 CD16hi NK(CD8-) | 16,246 | | | | 15,794 | | | | | | | | | | | | 33 | | | | | | | | | | 413 |
| 13 CD16hi NK(CD8-) | 11,569 | | | | 11,328 | | | | | 1 | | | | | | | 9 | 1 | | | | | | | | 1 | 225 |
| 14 CD16hi NK(CD8-) | 5,374 | | | | 5,146 | | | | | | | | | | | | 32 | | | | | | | | | | 193 |
| 15 CD16hi NK(CD161-) | 4,058 | | | | 3,912 | | | | | | | | | | | | 29 | | | | | | | | | | 116 |
| 16 CD16hi NK(CD161-) | 2,924 | | | | 2,584 | | | | | | | | | | | | 51 | | | | | | | | | 1 | 287 |
| 17 CD4 EM + CD4 CM(CD49d_alpha4) | 39,316 | | | 2,740 | | 17,755 | 14,985 | | | | | | | 24 | 5 | | | | | | | 1,073 | 15 | | | | 1,319 |
| 18 CD4 EM(CD25-) | 11,757 | | | 190 | | 7,404 | 3,372 | | | | | | | 8 | | | | | | | | 279 | 3 | | | | 307 |
| 19 CD4 EM(CD8-) | 5,934 | | | 3 | | 4,124 | 25 | | | 1 | | | | 5 | | | | | | | | 1,068 | 3 | | | | 655 |
| 20 CD4 EM(CD25- Foxp3-) | 4,040 | | | | | 3,742 | | | | | | | | 1 | | | | | | | | 106 | | | | | 175 |
| 21 CD4 EM(CD25-) | 5,124 | | | 92 | | 2,995 | 1,806 | | | | | | | 12 | 1 | | | | | | | 101 | 1 | | | | 114 |
| 22 CD4 EM(CD49d_alpha4-) | 2,263 | | | 1 | | 1,725 | 116 | | | | | | | | 1 | | | | | | | 81 | | | | | 244 |
| 23 CD4 EM(CD161-) | 1,935 | | | 4 | | 1,556 | 93 | | | | | | | | | | | | | | | 125 | | | | | 99 |
| 37 CD4 CM(CCR7-) | 8,456 | | | 471 | | 758 | 6,554 | | | | | | | | | | | | | | | 88 | | | | | 251 |
| 38 CD4 CM(CCR7-) | 3,537 | | | 258 | | 1,008 | 2,054 | | | | | | | | | | | | | | | 105 | | | | | 107 |
| 29 CD8 Temra(CD45RA+) | 11,637 | | | | 8 | | | 10,572 | 3 | 609 | | | 2 | 11 | | | | 1 | | | | 3 | | | 2 | | 426 |
| 30 CD8 Temra(CD161- Va7.2-) | 10,907 | | | | 1 | | | 9,354 | 4 | 995 | | | 3 | 2 | | | | | | | | 2 | | | 3 | | 469 |
| 31 CD8 Temra(Va7.2-) | 6,366 | | | | | | | 6,095 | | | | | | 30 | | | | | | | | | | | | | 222 |
| 39 CD8 N(CCR7-) | 25,334 | | | | | | | 1,393 | 22,975 | | | | | 12 | 3 | | | 3 | 1 | | | 2 | | | 663 | | 275 |
| 42 CD8 EM + CD8 CM(CD49d_alpha4) | 14,277 | | | | | | | 570 | | 10,123 | | | | 5 | | | | | | | | 3 | | | 1,277 | | 2,282 |
| 43 CD8 EM(CD49d_alpha4) | 3,666 | | | | | | | | | 2,796 | | | | 4 | 3 | | | | | | | 1 | | | 3 | | 828 |
| 49 Naive B(CD38-) | 10,845 | | | | 2 | | | | | | 10,651 | 2 | | | | | | | 100 | 2 | | | | | | | 45 |
| 50 Naive B(CD38-) | 1,596 | | | | 1 | | | | | | 1,368 | 1 | | | | | | | 183 | | | 1 | | | | | 21 |
| 51 Naive B(CCR7 CD38+) | 893 | | | | | | | | | | 847 | | | | | | | | 2 | | | 1 | | | | | 31 |
| 53 Basophils(CD25-) | 8,794 | | | 2 | | | | | | | | 8,688 | | | | | | | 1 | 1 | | 7 | | | | | 44 |
| 54 Basophils(CD25-) | 3,051 | 3 | | 1 | | | | | | | | 2,952 | | | | | | | | 5 | | 1 | 2 | | | | 45 |
| 56 CD16hi Mono(CD66- CD123-) | 8,662 | | | | 3 | | | | | | | | 8,365 | | | | 52 | | 1 | 1 | | | | | | | 207 |
| 57 CD16hi Mono(CD66- CD123-) | 3,596 | | | 10 | 3 | | | | | 1 | | 2 | 3,373 | | | | 39 | | | 1 | | | | | | | 133 |
| 58 CD8 MAIT(CD38- Va7.2-) | 5,061 | | | | 1 | | | 101 | 25 | 99 | | | | 4,472 | 8 | | | | | | | 1 | | | | | 354 |
| 59 CD8 MAIT(Va7.2-) | 3,978 | | | | | | | 2 | | 14 | | | | 3,676 | 1 | | | | | | | 1 | | | | | 284 |
| 63 gd Tcells(CD161) | 3,152 | | | | | | | | | | | | | | 3,135 | | | | | | | | | | | | 17 |
| 64 gd Tcells(CD8- CD11c-) | 2,954 | | | | | | 5 | | 4 | | | | | | 2,554 | 1 | | | | | | | | | | | 390 |
| 65 gd Tcells(CD161-) | 1,689 | | | | | | | | | | | | | | 1,655 | | | | | | | | | | | | 34 |
| 66 gd Tcells(CD56-) | 1,178 | | | | | | | | | | | | | | 992 | 1 | | | | | | | | | | | 170 |
| 68 Neutrophil(CD14- CD49d_alpha4) | 8,420 | | | 32 | | 24 | | | | | | | | 6 | | 7,880 | | | | | | | 13 | | | | 353 |
| 69 int Mono(CD38- HLADR+) | 5,623 | | | 4 | | 1 | | | | | | | | | | | 5,458 | 1 | | 1 | | | | | | | 149 |
| 70 int Mono(CD56-) | 2,320 | | | 358 | | | | | | 2 | | | | 12 | 1,666 | | | | 1 | | | | | | | | 277 |
| 72 CD56hi NK(CD16- CD56+) | 3,677 | | | | | | 1 | | | | | | | | | | | 2,972 | 2 | 1 | | 2 | | | 1 | | 696 |
| 73 Mem B(CD25-) | 1,868 | | | | | | | | | 2 | | | | | | | | | 1,658 | 53 | | | | | 30 | | 65 |
| 74 Mem B(CD25-) | 1,353 | | | | | 1 | | | | 1 | | | | | | | | | 1,185 | 14 | | 1 | 1 | | 54 | | 50 |
| 75 Trans B(CD25-) | 2,199 | | | | | | | | | | | | | | | | | | | 2,131 | | | | | 7 | | 47 |
| 76 Trans B(CD25-) | 687 | | | | | | | | | | | | | | | | | | | 662 | | | | | 4 | | 20 |
| 77 pDCs(FcERI-) | 1,582 | | | 3 | | | | | 7 | | | | | | | | | 1 | | | 1,543 | 1 | | | | | 31 |
| 78 pDCs(FcERI-) | 1,054 | | | 3 | 2 | | | | | 8 | | | | | | | | | 1 | | 954 | 1 | | | | | 41 |
| 79 CD4 Temra(CCR7-) | 618 | | | | | | | | | | | | | 7 | | | | | | | | 538 | 1 | | | | 29 |
| 80 Eosinophil(CD66+) | 2,287 | | | | | | | | | | | | | | | | 1 | | | | | | 2,176 | | | 1 | 108 |
| 87 mDCs(FcERI) | 3,422 | 1,154 | | | 1 | 1 | 1 | | | | | | | | | | | | | 1 | 2 | 2 | | 1,766 | | 1 | 488 |
| 89 Plasmablast(CD11c- CD27+) | 1,749 | | | 1 | | | | | | | | | | | | | | 117 | 16 | 2 | | | | | | 1,242 | 113 |

**Fig. 6 | Eshghi match table-selected.** Eshghi Match Table focusing on larger populations including 24 reference populations and 51 EPP phenotypes. The Column headers show the manually gated Reference populations and their event numbers. The Rows show the EPP-identified populations with cell type names from their matched Reference populations and indicators of markers used in the EPP phenotype. The associated +/- levels are based on population positions in the EPP graphs. The main table cells show the number of shared events between the corresponding Reference and EPP populations. GREEN highlights indicate dominant matches. The ORANGE cells indicate a second Reference population that contains part of an EPP phenotype.

phenotype. Comparing the Central Similarity and Jaccard Similarity coefficients indicates the extent of discrepancies between the reference and EPP event inclusion in the outer ranges of the distributions.

When more than one EPP leaf aligns to a reference phenotype, we treat the EPP leaves as components of the reference phenotype and combine them for the comparison. For about 1/3 of the reference phenotypes, there is a one-to-one correspondence between a reference population and an EPP leaf. Most of the rest of the reference phenotypes are split in EPP so that multiple EPP leaves cover parts of each reference population.

Overall, the Jaccard Similarity and Central Similarity results show almost perfect matching of reference and EPP phenotypes except where the reference gating divides populations in places where there is no clear indication of separation. When multiple EPP leaves that match a reference phenotype are taken together, about 2/3 of the Similarity and Central Similarity coefficients are over 80% and 88%, respectively. The consistent result across the Similarity table is that the Central Similarity values are always higher than the Jaccard similarities. This is the result expected, so long as the populations being compared represent the same basic phenotype. As noted in the individual reference comparisons above, some reference gating decisions seem to be based on expected marker expression for known phenotypes resulting in gating boundaries where distinct populations are not visible in the data. In several cases a single EPP leaf incorporates most of two reference populations or one or several EPP leaves include parts of two reference populations that are not well resolved in the reference gating. Since EPP is entirely data driven, it fails to separate such populations. When reference gates are drawn in well-populated areas, EPP may find multiple phenotypes aligned with one side or the other and also some straddling the division. In the Eshghi

example, for the Ref 11 CD4 CM and Ref 12 CD4 EM populations, EPP found several phenotypes that aligned predominantly with one or the other of these, but the most populous EPP phenotype constituting almost half of the total was evenly divided between them (Fig. 6, Row label 17). EPP also identified some distinct low-frequency phenotypes consisting of doublets that were included in spite of doublet exclusion gating (noted in the OMIP-044 section 2.3.2). Since the effectiveness of doublet exclusion varies between instruments and samples, identifying these populations may or may not be important.

## Discussion

Since the EPP gating sequence is composed of a sequence of ordinary two-dimensional projection steps, the results can be readily reviewed and understood using ordinary cytometry software, and the points in the analysis can be edited. Further analysis of the identified populations can be carried forward using multidimensional methods and including data dimensions that were not expected to identify discrete phenotypes, and thus were not used by EPP. In some cases, like cytokine staining, clear separation is not expected although the measurement distribution extends to clearly positive levels. In this situation, gate setting using FMO distributions may be the best way to define positive events more objectively, but EPP algorithmic phenotypes should provide the best available starting point for subsequent analysis.

The goal of EPP is to provide comprehensive discovery of all well-supported phenotypes in the data. In practice, the identification of valid separations will be affected by the choice of data transforms, test of statistical significance and the boundary score used, but within that limitation, EPP offers confidence that the analysis is exhaustive. A benefit of exhaustive

| Publication | Published Phenotype(s) | Similarity (Jaccard index) | Central similarity | EPP Phenotypes in comparison | Reference Populations in comparison | Reference Population Frequency | Number of Reference events |
|---|---|---|---|---|---|---|---|
| OMIP-047 | IgA+ | 83.1% | 89.7% | 1 | 1 | 5.1% | 2,746 |
| OMIP-047 | IgG1+ | 80.4% | 88.8% | 1 | 1 | 3.3% | 1,788 |
| OMIP-047 | IgG3+ | 87.9% | 95.9% | 2 | 1 | 0.7% | 377 |
| **OMIP-047** | **MZ B + Non-IgA_IgG1_IgG3*** | **86.4%** | **90.8%** | **1** | **2** | **39.0%** | **20,850** |
| **OMIP-047** | **Naïve & transitional*** | **90.1%** | **95.7%** | **1** | **2** | **44.9%** | **24,002** |
| OMIP-047 | PB | 55.2% | 65.0% | 1 | 1 | 0.4% | 215 |
| OMIP−044 | [not DCs] | 15.3% | 10.3% | 2 | 1 | 0.8% | 16,476 |
| OMIP−044 | [not naive CD4+] | 85.3% | 88.7% | 5 | 1 | 17.2% | 339,632 |
| OMIP−044 | [not naive CD8+] | 90.8% | 93.9% | 3 | 1 | 5.4% | 105,744 |
| OMIP−044 | CD123+ pDC | 96.3% | 99.6% | 2 | 1 | 1.2% | 23,462 |
| OMIP−044 | CD14+ | 97.2% | 100.0% | 3 | 1 | 18.8% | 371,366 |
| OMIP−044 | CD141+DCs | 46.2% | 51.2% | 1 | 1 | 0.05% | 908 |
| OMIP−044 | CD19+ | 99.3% | 100.0% | 2 | 1 | 12.8% | 252,670 |
| OMIP−044 | CD1c DCs | 63.0% | 68.7% | 1 | 1 | 0.4% | 8,222 |
| OMIP−044 | CD56+ NK | 95.3% | 99.6% | 6 | 1 | 7.3% | 144,124 |
| OMIP−044 | DN | 83.8% | 88.0% | 5 | 1 | 2.7% | 53,390 |
| OMIP−044 | DN DCs | 82.7% | 91.3% | 1 | 1 | 2.6% | 51,976 |
| OMIP−044 | naive CD4+ | 88.7% | 91.1% | 2 | 1 | 23.0% | 453,746 |
| OMIP−044 | naive CD8+ | 93.5% | 97.1% | 2 | 1 | 6.6% | 131,064 |
| OMIP-077 | **[NonMZB cells] & MZB cells*** | **95.9%** | **98.5%** | **1** | **2** | **1.9%** | **5,085** |
| OMIP-077 | Basophils | 94.6% | 99.3% | 1 | 1 | 0.3% | 889 |
| OMIP-077 | Eosinophils | 98.4% | 100.0% | 1 | 1 | 1.3% | 3,329 |
| OMIP-077 | Monocytes | 94.0% | 96.6% | 3 | 1 | 6.9% | 18,228 |
| OMIP-077 | **mDCs*** | **55.0%** | **59.4%** | **1** | **3** | **0.1%** | **213** |
| OMIP-077 | Neutrophils | 99.6% | 100.0% | 3 | 1 | 76.5% | 201,691 |
| OMIP-077 | NKcells | 90.9% | 99.2% | 2 | 1 | 2.0% | 5,149 |
| OMIP-077 | pDCs | 97.4% | 100.0% | 1 | 1 | 0.04% | 112 |
| OMIP-077 | Plasma cells | 91.6% | 96.3% | 1 | 1 | 0.1% | 227 |
| OMIP-077 | Progenitors | 74.9% | 87.8% | 1 | 1 | 0.1% | 181 |
| OMIP-077 | T cells | 99.7% | 100.0% | 2 | 1 | 10.6% | 27,847 |
| Eshghi | [CD1c- HLADR-] | 27.4% | 30.5% | 1 | 1 | 0.1% | 338 |
| Eshghi | Basophils | 97.6% | 99.6% | 3 | 1 | 2.4% | 11,906 |
| Eshghi | CD16hi Mono | 95.5% | 98.3% | 2 | 1 | 2.4% | 11,776 |
| Eshghi | CD16hi NK | 95.7% | 99.0% | 6 | 1 | 11.3% | 56,449 |
| Eshghi | CD4 CM | 20.7% | 22.7% | 1 | 1 | 5.9% | 29,702 |
| Eshghi | CD4 EM | 53.8% | 55.8% | 9 | 1 | 9.1% | 45,354 |
| Eshghi | CD4 N | 85.8% | 93.2% | 3 | 1 | 14.0% | 70,143 |
| Eshghi | CD4 Temra | 13.1% | 12.6% | 6 | 1 | 1.6% | 8,120 |
| Eshghi | CD56hi NK | 74.2% | 79.7% | 1 | 1 | 0.7% | 3,298 |
| Eshghi | CD8 CM | 7.3% | 7.3% | 1 | 1 | 0.5% | 2,266 |
| Eshghi | CD8 EM | 62.7% | 66.1% | 7 | 1 | 3.4% | 16,756 |
| Eshghi | CD8 MAIT | 89.6% | 93.7% | 5 | 1 | 1.8% | 8,886 |
| Eshghi | CD8 N | 89.5% | 95.3% | 3 | 1 | 4.7% | 23,383 |
| Eshghi | CD8 Temra | 82.2% | 86.8% | 5 | 1 | 6.0% | 29,765 |
| Eshghi | Eosinophil | 86.3% | 95.3% | 1 | 1 | 0.5% | 2,409 |
| Eshghi | gd Tcells | 90.1% | 96.1% | 5 | 1 | 1.8% | 9,049 |
| Eshghi | IgD- CD27- B | 54.5% | 63.9% | 1 | 1 | 0.2% | 858 |
| Eshghi | int Mono | 80.7% | 85.0% | 3 | 1 | 1.6% | 8,126 |
| Eshghi | Lineage Negative | 42.2% | 48.6% | 4 | 1 | 0.5% | 2,729 |
| Eshghi | mDCs | 51.1% | 54.6% | 2 | 1 | 0.4% | 1,833 |
| Eshghi | Mem B | 83.2% | 91.2% | 2 | 1 | 0.6% | 3,041 |
| Eshghi | **HLADRio + CD14hi Mono*** | **93.2%** | **97.0%** | **1** | **2** | **18.4%** | **92,060** |
| Eshghi | Naive B | 95.9% | 99.4% | 4 | 1 | 2.6% | 13,016 |
| Eshghi | Neutrophil | 93.0% | 98.0% | 1 | 1 | 1.6% | 7,929 |
| Eshghi | pDCs | 93.6% | 99.2% | 2 | 1 | 0.5% | 2,519 |
| Eshghi | Plasmablast | 66.9% | 79.6% | 1 | 1 | 0.3% | 1,350 |
| Eshghi | Trans B | 85.4% | 89.0% | 2 | 1 | 0.6% | 3,179 |
| Eshghi | Treg | 37.8% | 44.1% | 1 | 1 | 1.0% | 4,833 |

**Fig. 7 | EPP-reference similarity table.** Composite similarity table comparing major populations in the four reference publications with the aggregate of the events in matching EPP phenotypes. In several cases (marked by *), reference gating boundaries did not indicate any obvious separation of populations and EPP identified only a single corresponding leaf. In these cases, the reference populations were combined for similarity comparison to the EPP leaf.

analysis is that EPP will identify unexpected phenotypes and splits of known populations, which can be investigated.

Since EPP makes no assumptions about the cell populations, it avoids the pitfalls of confirmation bias. Instead, always seeking the separations between populations with the least overlap should produce the analysis with the greatest biological fidelity. The algorithmic phenotypes are the best representation of the biological phenotypes achievable with the data, i.e., the ones with the fewest misallocated cells.

The reference analyses serve two purposes. In the Results we describe where EPP agrees with the reference analyses and where it doesn't and why. Making EPP phenotypes match reference populations as closely as possible was not an objective of our work. The reference analyses also provide the biological context needed to interpret the EPP results.

The previous approaches that have been best received by biologists, flowDensity[15] and openCyto[16] are toolkits for automating gating strategies predefined by experts. Most other approaches, for example flowClust[17], do not directly produce gating structures suitable for sorting instruments and for many, such as FlowSOM[18] or SPADE[19], this is infeasible. The methods used are diverse but most can be posed as optimization problems and the merit function used by EPP is firmly grounded in practical biology. The preferred solutions are those for which clean biological verification is possible because the populations are well separated, i.e., they could be sorted with good purity. A second important aspect of each algorithm is determining the number of clusters, which in the case of hierarchical methods, as here, requires an appropriate stopping criterion. EPP is unique, as far as we know, in using a rigorous statistical test[10] to achieve this, and this is only possible because we preserve real density estimators throughout. For example, t-SNE[7] or UMAP[20] plots do not do this, so a similar test is not possible.

A recent publication on an unsupervised clustering technique using sequential projection pursuit termed APP by Simpson, et al.[21] shares some features with EPP but is based on entirely different computational methods and evaluation measures. Since it uses a sequence of two- dimensional projections, APP results should share with EPP the advantage of providing a transparent analysis sequence. However, although the necessary steps in projection pursuit analysis are similar in EPP and APP, the methods used at each stage (e.g., identifying candidate splits, scoring candidate splits and deciding to stop when a population has no valid splits) are unrelated. We see the use by EPP of a statistical test as a stopping criterion as a benefit compared to the method used in APP.

By design, EPP employs the same strategies for spatial reasoning as biologists do, independent of the details of the biology beyond the assumption that the cell data are a sample representing a collection of phenotypes. EPP uses already familiar tools, albeit in novel ways, and the EPP algorithmic phenotypes can serve as the starting point for biological analysis instead of the individual cell data. The existence of a trustworthy agent, like EPP, to do the first reduction of cell data to valid populations for biologists to review and interpret according to biological criteria, could break a long-standing logjam in cytometry informatics.

## Methods
### Data normalization
Fluorescence data were compensated as defined in their repository submissions and transformed onto the interval [0, 1] using a 4.5 decade logicle scale with width $W$ chosen to maintain data of interest on scale without excessive compression around data zero (Width range 0.5 to 1.55, most commonly 0.8-1.1). Mass cytometry data were similarly transformed using a 4.3 decade, $W = 0$ logicle transform, which is arcsinh rescaled to align the top of scale with a conventional value and the small integer values mapped into the first "decade". For each data set, preliminary gating was carried out in FlowJo to isolate live single cells of interest, guided by the gating sequence shown in the publication. The resulting set of events was submitted to the EPP process. To avoid the overhead of bounds checking in the inner loop, any observation with a data value outside the closed interval [0, 1] is censored.

Staining panels often include some markers intended for phenotyping and others used for evaluation of the identified phenotypes. Since the latter may not provide separable measurement distributions, omitting them from the EPP analysis may simplify and speed up the process. For our comparisons to the published phenotyping results, we used the same set of data dimensions for EPP that were used to obtain the reference phenotypes.

### EPP implementation
To improve the efficiency of the computation, EPP initially tests each data dimension of the current population to see if it is likely to be useful, and only the qualified dimensions are examined further to identify the best available separation (see Fig. 1) For the initial population and all subsequent sub-populations, the EPP process examines two-dimensional data distributions for all pairs of these dimensions to identify the best way to make a two-way split in the data. Once a split has been selected, EPP is applied recursively to each branch until no additional splits are found. At that point, the current population is designated as a "leaf" or algorithmic phenotype.

The computation is divided into discrete units, which can be executed safely in different threads. The default is to use the number of cores the processor declares. Starting with the whole sample, for each population considered, a qualification step is queued for each dimension marked as being suitable for phenotyping. This first estimates the mean and variance of each single dimension and sorts the data. These sorted data are the order statistics of the sample distribution and from them and the cumulative distribution function of the normal and exponential distributions having the same estimated mean and variance, we compute the Kullback-Leibler Divergence (KLD)[9] of the sample distribution from the parametric distributions. Any dimension for which the divergence from normal or exponential is below the corresponding threshold is omitted from further analysis of this population. In this way, dimensions in the current population that appear to have a uniform positive or negative stain are omitted. For the normal distribution, a threshold of .04 has been used for the evaluations presented here. For the exponential distribution, a threshold of .2 was used. Every dimension that qualifies is paired with each of the previously qualified dimensions, the pair queued for the next step and the dimension added to the qualified list. If only one dimension qualifies, it is paired with the unqualified dimension with the highest divergence from normal, and if none qualify, then the two with the highest divergence from normal are paired, so that in every case, at least the most promising pair of dimensions is tried.

For each pair of the qualifying dimensions, we need an estimate of the sample distribution function or "density". First compute a weight array from the provided data values of the population for the selected dimensions, as described in DBM[10]. A grid size of $257 \times 257$ is used based on our prior work. A Gaussian kernel density estimator based on the same grid is then computed using a discrete cosine transform (DCT)[22,23] implemented by the FFTW package[24]. The algorithm is adaptive; if a solution can't be found at the highest resolution, the kernel width $W$ is increased by a factor of $\sqrt{2}$, i.e., the bandwidth is reduced, and the computation restarted. If necessary, this is repeated. On the fluorescence data sets, the initial standard deviation of the Gaussian kernel was set to W = .01 or 1% of full scale. Much less than this is not appropriate for a 257 grid, accounting for the additional bandwidth needed by DBM. On our one mass cytometry data set, we found W = .025 to behave more stably. With these settings only a small percentage of cases required more than one pass. Processing time is linear in the number of observations and the time to compute the density estimator is constant, although in practice, it will be multiplied by the average number of passes.

The next step is modal clustering on the estimated density. First sort the grid points by density. Next establish the lower limit on the density required to found a new cluster, to limit the effect of counting noise in sparse regions. Compute the number of grid squares with an area close to the spot size $\pm 2W$ of the current kernel and move a window of this width along the sorted grid points, from lowest density upward until the total weight within the window exceeds $\sigma^2$, $\sigma = 3$ was used here and 9 events $\pm 2W$ are required. Then for each grid point, in descending order of density down to that

threshold, examine the four closest neighbors. If none have been assigned to a cluster, then we have identified a local maximum or mode, so start a new cluster containing the point. If two neighbors belong to different clusters, mark it as a boundary point. Otherwise, the point is added to its neighbor's cluster. Seven of the eight contiguous neighbors are marked as being contiguous with the classified points, but one of the diagonal ones is omitted based on a counter modulo 4, in order to make this operation more radially symmetric. For the remainder of grid points with density below threshold, find all the points marked contiguous with the classified set that are not already classified, classify them as above and repeat until all are classified. This assigns these points to the nearest significant cluster. If there are too many clusters for the implementation to handle in reasonable time, 12 was used here, widen the kernel by a factor of $\sqrt{2}$ and start over again from the weights, if necessary, repeating this process. This was chosen empirically as the point where it begins to be faster to do the additional FFT than spend more time graph processing. The choice of $\sqrt{2}$ saves time, since one of the two DCT needed for DBM can be reused. Optionally the process is also stopped when the absolute or relative population size falls below a specified value. Running time is not constant but is only weakly influenced by the actual data.

If some separation was found, the boundary points are each examined to determine which of their neighbors is also a boundary point and this defines one or more segments of an edge. All segments found in this way are then linked end to end, using an ordered map to search the range of feasible neighbors, which reduces the computational complexity from $O(n^2)$ to $O(n \log(n))$, where $n$ is the number of segments. This produces a plane graph where each cluster is contained in one face of the graph, the edges of the graph are the linked boundary segments between two clusters and the graph vertices are the points where three or more clusters meet. There are a few pathological graphs, with edges that cannot be consistently assigned, and the boundary is not of measure zero, which must be handled specially, but they only occur in very complex graphs and are extremely rare in practice and this has no effect on performance. If the cluster graph is too complex for the implementation to process, widen the kernel and restart from the weights as above. Our implementation is 32 bits, allowing no more than 32 edges but graphs requiring more with 12 nodes or fewer, while theoretically possible, are extremely rare in our experience. If no clusters are found, the pair of dimensions is dropped.

The graph is used to find the cluster assigned to each event and to compute the weights that will be used for scoring the boundaries. Using the imposed order, a constant time lookup table based on one dimension is used to locate the first feasible edge segment, followed by a linear search until an edge is found or all feasible segments have been excluded and cluster membership assigned, giving linear time in the number of segments and in the number of observations.

Density Based Merging is now done by first finding the highest density point along each edge, and then in descending order, testing that the densities at the two local maxima separated by the edge are significantly different than that of the edge. If not, remove the edge and restart the process. This requires a second DCT computation, which however, is equal to the prior pass density estimator when the width increases by $\sqrt{2}$, in which case, it only needs to be performed on the first pass. The test is based on statistics derived via DCT[22], not direct calculation, but otherwise is as described in DBM[10].

If more than one edge remains, then we must find the subsets of edges that are suitable separation boundaries, i.e., continuous and dividing two contiguous regions. The dual of a simplified dual graph however, always has these properties. Therefore, we can do this by systematically removing one dual edge at a time and merging two dual vertices or nodes. Removing a dual edge means that two nodes (clusters) are merged and may cause other dual edges to merge. Since the dual graph is abstract, dual edges and dual vertices (nodes) can be represented as positions in bit vectors and then the operation of merging two edges or nodes is simply Boolean arithmetic. Since we don't care what order the edges are removed, we can use the bit order and need only consider the case where edges are removed in this order. Without the ordering this would be an $O(n!)$ process, where $n$ is the number of faces of

the cluster graph. Ordering makes this closer to a binomial coefficient $O(C(n, k))$ with $k$ set by the effective number of faces (clusters) that are merged to form simple graphs. A binary representation of the dual graph of the cluster graph is pushed onto a stack. While this stack is not empty, the top element is removed and examined. If it is simple, i.e., has only one edge remaining then it is the dual of a candidate boundary, which is passed to the next step. If not, examine each dual edge contained in this graph and, if it is greater than all the previously removed edges of the graph, push a simplified dual graph with that edge removed onto the stack.

A score is computed for each candidate separation by integrating the density along the boundary. This is the "best separation" mode. When "best balance" mode is requested (as in the results presented here), starting with the cluster weights, the fraction of events in each subset is computed as $P$ and $(1 - P)$, and the score is divided by a balance factor $4P(1 - P)$, which penalizes unbalanced splits. When all outstanding computations are complete and the candidates have been scored, the one with the best score (if any) is chosen to define two new subpopulations in the gating tree and the process is restarted on these. When no new separations are identified among the current populations, the EPP process is complete, and these populations represent the algorithmic phenotypes.

The resulting gating tree is JSON[12] encoded. By default, polygons are simplified to a tolerance of 1% using the Ramer-Douglas-Peucker algorithm[25,26]. The JSON includes the scores for each gating tree boundary, as the estimated number of events within a strip $\pm W$ wide along the border but the scaling is only approximate. A separate CSV file containing an assigned numerical cluster identifier for each observation as well as the Mahalanobis distance of the observation from the cluster center is also produced, if requested. When called from our MATLAB utility, EPP's JSON output is used to augment a workspace created by FlowJo, so that the EPP populations and gates can be displayed and used as in any other FlowJo gating tree.

## Statistics and reproducibility
Our implementation is deterministic. However, the algorithm uses pseudo-random techniques and returns polygons accurate to a specified tolerance, which must be chosen appropriately for the application. We use only descriptive statistics.

## Reporting summary
Further information on research design is available in the Nature Portfolio Reporting Summary linked to this article.

## Data availability
The data sets used in this paper are available from the ISAC Flow Repository. **OMIP-047** http://flowrepository.org/id/FR-FCM-ZYFB **OMIP-077** http://flowrepository.org/id/FR-FCM-Z3HC **ESHGHI** http://flowrepository.org/id/FR-FCM-Z24F **OMIP-044** http://flowrepository.org/id/FR-FCM-ZYC2 FlowJo Workspaces containing our recreation of the published gating and the EPP analyses described here are available for download. **OMIP-044** https://storage.googleapis.com/cytogenie.org/EPP/WSP/2024_11_05/240915%20Fig8-9-10%20OMIP-044%20DRP6b.wsp **OMIP-047** https://storage.googleapis.com/cytogenie.org/EPP/WSP/2024_11_05/240915%20Fig12%20OMIP-047%20DRP4b.wsp **OMIP-077** https://storage.googleapis.com/cytogenie.org/EPP/WSP/2024_11_05/240915%20Fig1%267%20OMIP-077%20DRP3y.wsp **ESHGHI** https://storage.googleapis.com/cytogenie.org/EPP/WSP/2024_11_05/240915%20Fig13%2614%20ESHGHI%20DRP2gm.wsp.

## Code availability
The C++ source code is available from GitHub https://github.com/black98fxstc/epp-swarm/tree/v1.0.0 under the BSD license. CMAKE build scripts are provided for Windows, Mac and Linux and Clang, MSVC and Gnu compilers are supported. The code is dependent on the FFTW Fourier transform package http://fftw.org/ and the NLohmann JSON implementation https://github.com/nlohmann/json both of which are freely available. The

compiled code can be linked to other programs and called directly to analyze data in memory. A simple command line utility is provided that uses CSV and JSON files as input and output. For those not wishing to do software development a repackaged version is available that may be more suitable https://storage.googleapis.com/cytogenie/EPP/build.zip with documentation https://docs.google.com/document/d/1So9JHOja7iBncJHnIp2KoYBVvh5_ZjUaLqKwbR5am8Y/edit?usp=sharing. The MATLAB/FlowJo bridge is documented here https://docs.google.com/document/d/1sFuTIf6iadisiOiQClZ1e_lywppL8rt09RsHp2JKYuI/edit?usp=sharing and source code is available here https://storage.googleapis.com/cytogenie/GetDown2/domains/FACS/AutoGate_v5.95.zip. It is also released under the BSD license. It does require a full MATLAB license to run, which is usually available to academic users.

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

## Acknowledgements

The authors thank Darya Orlova (currently at Cellazon) and her colleagues at Genentech for access to the mass cytometry data sets and Eliver Ghosn for prototyping data and discussions in the early part of the project. The authors also thank Josef Spidlen and Dylan Hinson from Becton-Dickinson FlowJo for providing software development support and licenses. This work was supported by grants (AI 098519) from the National Institutes of Health (US) and (DMS-1916074) from the National Science Foundation (US).

## Author contributions

W.M., D.P., S.M., G.W. and L.A.H. contributed to the conception and design. S.M. was responsible for the development software implementation in MATLAB that validated the method, and the bridge software with aid of C.M. and W.M. W.M. is responsible for the re-implementation in C++, with support from S.M., C.M. and G.W. W.M. drafted the methods and D.P. drafted the data analysis for the article with help from C.M. All authors critically revised the article for important intellectual content.

## Competing interests

The authors declare no competing interests.
