## [Transparent Peer Review file · Communications Biology]

Automatic Phenotyping Using Exhaustive Projection Pursuit

Corresponding Author: Mr Wayne Moore

Version 0:

Reviewer comments:

Reviewer #1

(Remarks to the Author)

A concern in cytometry data analysis is the potential introduction of bias in the definition of population hierarchies and phenotypes via manual approaches. Bias can derive from the chosen order of parameter pairs and the definition of population boundaries via manual gates. Both will affect the quality of the analysis, and the results derived from it, including phenotype misclassification and biologically relevant subsets being missed or neglected during the analysis. The growing complexity of polychromatic panels nowadays and the recent introduction of spectral analyzers able to measure 45+ fluorescence labelled markers per sample, further compounds the issue of bias and the impracticality of manual analysis approaches (lengthy, assessed via direct visualization without robust metrics, and thus affected by analyst's bias and error), calling for better ways to analyze biological data in a quick yet exhaustive, data-driven fashion.

This manuscript covers the development and assessment of an automated tool (Exhaustive Projection Pursuit or EPP) for the analysis of cytometry data and the comprehensive identification of distinct cell phenotypes present in heterogeneous samples. To achieve this, EPP evaluates data across all possible 2D combinations of selected relevant phenotype markers to identify the pair enabling the best initial splitting of events into distinct populations with defined boundaries. Gated events are subsequently re-analyzed across all parameter pairs to identify again the one pair allowing best resolution and a further splitting of events via gates. As it progresses, EPP progressively generates a sequence of analysis and a hierarchy of gates leading to the exhaustive identification of statistically relevant subpopulations (or "leaves"), which can't be further subdivided by the algorithm in any of the considered parameter pairs but can be further analyzed for the expression of markers excluded from the initial EPP-evaluation. EPP source code is freely available, results can be visualized via third party software and gates can be edited to adjust boundaries if necessary.

One of the advantages of EPP is its data-driven nature: Steps in the EPP algorithm are strictly based on data resolution; the order of parameter pairs considered in the gating strategy generated by EPP follows -for a given data set - the path where the best 2D resolution of phenotypically relevant subsets can be achieved. Although the algorithm's focus on the analysis of data resolution across parameters pairs in a 2-dimensional space may seem limiting at first, compared to the data mining power of high dimensionality reduction and clustering algorithms, this practical sequential 2D data analysis is still the procedure of choice by most cytometry analysts and the sole choice for most instrument-dedicated cytometry software. Sequential 2D analysis is used in cell sorters for the definition of sort gates, and it is also crucial to interpret and QC the phenotype of clusters generated via high dimensionality reduction and clustering algorithms. A preliminary thorough exploration of samples at the beginning of acquisition and using EPP, will also allow the adjustment of recording limits not to miss biological relevant subsets identified by the algorithm, enable robust statistics and clear definition of key cellular players in deep phenotyping and subset discovery projects.

Although readers lacking a background on programming or bioinformatics may find it difficult to understand the technical aspects of the EPP workflow and the metrics thoroughly described by the authors in the manuscript, they will be familiar with the approach taken by the algorithm to achieve a fast dissection of complex heterogeneous samples into unique subsets. As the authors mentioned in the last paragraph of the Discussion section (page 17), "EPP employs the same strategy for spatial reasoning as biologists do", only quicker and devoid of assumptions on the biology of samples, except for their intrinsic subset heterogeneity. Based on the comparison made by the authors between EPP's output and the manual analysis reported in OMIP 44, 47 and 77, the algorithm does a very good job in matching most of the reported reference phenotypes while identifying some that were excluded in the published studies. EPP could also highlight inconsistencies in phenotypes defined via manual approaches. EPP will be invaluable as a starting point for biological analysis and more: the possibility to integrate the algorithm's output into existing cytometry software will be particularly useful in cell sorting of very large

polychromatic samples and rare samples, where speed of analysis leading to the isolation of viable, fully functional cells is as important to the success of the sort as the quality of the sample and the design of the polychromatic panel. By removing assumptions on the expected subsets contained in the sample, EPP will also strengthen the chances to discover new unsuspected cellular phenotypes. I congratulate the authors for sharing a methodology for the automatic, unbiased analysis of complex heterogeneous samples that will certainly improve the robustness and speed of cytometry data analysis.

That said, I have a few minor suggestions to improve the experience of the reader:

All the figures seem tiny, with tinnier fonts for axis and gate labels, barely visible if at all in the A4 printed document. Increasing the fonts and the overall size of the plots will allow the user to assess and confirm comments made in the document regarding EPP outcome vs. published OMIP results. The same applies to the tables.

Figure legends and table descriptions are minimalist, providing a title but no further information enabling a clear understanding of the data presented, the meaning of columns or rows or the reason why some cells or fonts in the tables appear highlighted in green, blue, yellow, red or orange. Consider expanding the text to improve clarity or modifying the figures to remove unnecessary (distracting) highlights.

The name of some of the gates in the EPP figures are confusing. On Figure 1, 2nd column first row (c2:r1), subsets are labeled as "CD3-CD14+" and "CD3++CD14", instead of "CD3++CD14-". In the same figure (c3:r2) eosinophils are labelled as "Eosinophils CD19+". Isn't CD19 a B cell pan marker? I understand that, from an algorithm perspective, a population appearing brighter along the CD19 detector will be classified as positive for the sake of just tagging, even when the intensity is entirely due to autofluorescence and not the result of fluorescence labelling. This could raise a few eyebrows and confuse the reader not aware of this. Would it be wise to add a comment along the text on the reason for the labelling and the potential impact of AF on fake phenotype classification?

There's no information on the fluorochrome used to label each marker in the EPP results. Is this a limitation of the algorithm or parameter labels can be modified at will at some point along the workflow?

I couldn't find information in the text of the time taken by EPP complete, which I take will depend on the size of the panel and the number of markers selected for the C++ based analysis. This information is important to appreciate the practicality of the procedure.

Two final questions: given a defined cleaned data set, will EPP produce identical results across separate runs, or some variability could be expected? What would be the author's advice on the approach to be followed by researchers applying EPP to the simultaneous analysis to several samples differing in frequency and nature of subsets, including changes in marker expressions (naïve vs infected tissue samples)?

Reviewer #2

(Remarks to the Author)

Moore et al. proposed a data-driven gating strategy, Exhaustive Projection Pursuit (EPP), to divide the samples on the 2-markers panel into two parts based on the density of measurement distribution. The split boundary is very flexible compared to the horizontal and vertical manual gating. The authors also made several comparisons between their EPP data-driven gating and the manual gating from several case studies. The auto-gating algorithm seems to function very well. However, several major components are missing for a publishable paper and a convincing tool.

1. A model diagram and reader-friendly description of the logic of modeling are missing to make the EPP concept and procedure clear. The concept that the gating boundaries went through the region where the sample density on the 2-marker panel is lowest is clear. However, the algorithm and model to achieve this goal are not clearly explained. Section 2.1 is challenging to follow and filled with jargon without clear definitions, such as "face," "event," "dip," etc.

2. The restriction of strictly splitting each 2-marker panel into two sections can limit the flexibility and generalization of using EPP. In the manual gating practice, it is common to use three or four gates to define three to four cell populations. For example, CD3+/-CD19+/- for major separation of B, T and monocytes. It seems to me that after splitting CD3 vs CD19 panel into two sections, EPP will not gate on CD3 and CD19 again. This will cause inconvenience for labs that have organized gating strategies for their tissue systems. Can authors relax this limitation of bi-partition? Or allow another round of further bi-partition on the same marker pairs?

3. Authors compared EPP gating results with manual gating standards from each original publication of the case studies. However, it is not clear to me how the EPP gating sequence order is defined. It seems to me that EPP is a spatial splitting algorithm. However, in all of those EPP gating figures, such as Figure 1, how did EPP determine which pairs of markers to the gate first and within which of the two splitting sections, further gating on which another pair of markers to split?

A subsequent question is: did authors infer cell type and find the matching cell population with the manual gating results? For example, if a user is used to leverage CD3-/CD19+ to define B cell population. However, EPP split on CD3 vertically (CD3-, CD3+) and continues to gate CD3- population on other markers like CD14, CD16 etc, but not CD19 anymore. How

should this user define B cells? In other words, if the EPP gating sequence order does not follow the user's cell type annotation knowledge, how can the user find the matching cell populations split by EPP to define the cell type at the resolution they want correctly?

4. Another major missing component is the comparison with existing auto-gating strategies. Upon a quick search, it seems that multiple auto-gating software has been launched, such as flowClust and OpenCyto. There is even a benchmarking paper to assess all kinds of automated flow cytometry data analysis techniques more than 10 years ago:

<https://www.nature.com/articles/nmeth.2365>. Is anything novel and more powerful for EPP compared to all existing ones? I did not see any discussion in the current manuscript.

Reviewer #3

(Remarks to the Author)

Overall comment:

Moore et al. developed EPP (Exhaustive Projection Pursuit), a tool to automate gating strategies for cytometry data. The authors have done a nice comparison of EPP across four cytometry datasets to validate its performance. They successfully identified populations that were missed in the initial OMIP paper.

However, some sections can be more clearly described. It isn't immediately clear what the source of each gating strategy is, and some figures can be combined to improve the paper.

Despite these shortcomings, the paper has some exciting results that would be beneficial to the greater cytometry community. Great job!

Major comments:

- Create a figure that summarises how EPP works in section 2.1. There's a lot of information, but a figure would help clarify the workflow
- It's not always clear which gating strategy has been generated by manual gating or by EPP. This needs to be made clearer throughout

Minor comments:

- At the end of section 2.1, the runtime of EPP is mentioned but nothing is detailed. Include a table (can be supplementary) that quantifies the run time based on number of cells. Also include information on hardware (particularly if a HPC is used)
- Is Figure 1 generated using EPP or is it the gating strategy from the referenced paper? It's not immediately clear so re-word. It can also be combined with Figure 2
- Section 2.2 needs a figure to better visualise what is being described. It may be appropriate to combine with section 2.1, and/or include the mathematics/functions behind the described calculations
- In section 2.3.1, "Our emulation of the reference gating scheme is shown in Figure 7". Do you mean this was the original gating strategy as proposed by the OMIP or is this the result of EPP? It should be re-worded for clarity
- Figure 2: the EPP populations should be more clearly defined. Assuming Figure 1 contains the gating strategy developed by EPP, "7 Monocytes(CD16+)" are CD14++ whilst "Monocytes(CD16+CD19-)" are CD14-. These are intermediate and non-classical monocytes (respectively), whilst "6 Monocytes(CD16-)" (which are also CD14++) are classical monocytes. Better define the phenotypes of these populations
- Section 2.3.1 mentioned an opt-SNE plot in Figure 1 that doesn't exist
- Section 2.3.2: "The upper three rows in Figure 8 show our emulation of the main gating structure illustrated in Figure 1". This suggests Figure 8 shows the gating strategy of EPP, but I don't think that's what is trying to be said? It's also not clear how this section relates to figure 1 (OMIP-077). This paragraph should be re-worded as it's not clear what the message is
- Section 2.3.3: "Our emulation of the reference gating in Figure 1 of OMIP-047". Is "Figure 1" referring to the OMIP paper? If so, remove it (or call it something else) as it's confusing. Furthermore, if figures like Figure 11 are just copied from the OMIP paper they need to be correctly referenced/licensed
- Mass cytometry figures 14-17: adjust the axes so you can more clearly see the cells backed up against the axes
- T-SNE or UMAP plots would be useful for visualising the populations identified by EPP and how it compares to the reference OMIP dataset
- Make some of the figures supplementary
- Is it recommended to run EPP on a concatenated file of all samples (which may be skewed by batch effect) or to run each sample separately (which may in turn produce different results)?
- Most packages being used to analyse cytometry data are in R or Python. Making EPP available in these other languages will increase the usage. It's great that it's implemented in FlowJo

Version 1:

Reviewer comments:

Reviewer #1

(Remarks to the Author)

Dear Moore et. al,

Thank you for addressing all the questions and suggestions for improvements in my initial review.

Corrections of errors in the text and the addition of notes, together with the expansion of the figures legends to include relevant details have improved the clarity of the manuscript.

Thanks for adding the fluorochrome list in Supp Fig. 20 and for including information on the approximate duration of the procedure. I'm pleased with the new revised document.

Best of lucks in your future research!

Reviewer #2

(Remarks to the Author)

The authors have addressed my questions.

Reviewer #3

(Remarks to the Author)

My comments have been addressed, either through changes to the manuscript or through justification. I have not other comments.

Response to Reviewers for

Automatic Phenotyping Using Exhaustive Projection Pursuit

by W. Moore, S. Meehan, C. Meehan, D. Parks, G. Walther, L. Herzenberg

COMMSBIO-24-8103A

We thank the reviewers for their thoughtful and supportive evaluations and their effort in providing useful comments and criticisms. We have responded to each item in the following document and we think that we were able to substantially improve the manuscript in response. Places in the manuscript where changes have been made are marked with BLUE text and with Change Bars in the near right margin. In our responses to the reviewers' points that led to a change at a specific place in the text, we note the relevant line numbers.

We have generally retained the structure of the original submission to facilitate comparisons between reviewer comments and places where related revisions have been made. This has resulted in a few items that are not optimally aligned like the added "EPP Algorithm" figure illustrating the steps in the EPP process. To avoid renumbering previous figures, this new figure which might logically be Figure 1 appears as Figure 19 after the original Supplementary figures.

1 Reply to the Reviewer #1

A concern in cytometry data analysis is the potential introduction of bias in the definition of population hierarchies and phenotypes via manual approaches. Bias can derive from the chosen order of parameter pairs and the definition of population boundaries via manual gates. Both will affect the quality of the analysis, and the results derived from it, including phenotype misclassification and biologically relevant subsets being missed or neglected during the analysis. The growing complexity of polychromatic panels nowadays and the recent introduction of spectral analyzers able to measure 45+ fluorescence labelled markers per sample, further compounds the issue of bias and the impracticality of manual analysis approaches (lengthy, assessed via direct visualization without robust metrics, and thus affected by analyst's bias and error), calling for better ways to analyze biological data in a quick yet exhaustive, data-driven fashion.

This manuscript covers the development and assessment of an automated tool (Exhaustive Projection Pursuit or EPP) for the analysis of cytometry data and the comprehensive identification of distinct cell phenotypes present in heterogeneous samples. To achieve this, EPP evaluates data across all possible 2D combinations of selected relevant phenotype markers to identify the pair enabling the best initial splitting of events into distinct populations with defined boundaries. Gated events are subsequently re-analyzed across all parameter pairs to identify again the one pair allowing best resolution and a further splitting of events via gates. As it progresses, EPP progressively generates a sequence of analysis and a hierarchy of gates leading to the exhaustive identification of statistically relevant subpopulations (or "leaves"), which can't be further subdivided by the algorithm in any of the considered parameter pairs but can be further analyzed for the expression of markers excluded from the initial EPP-evaluation. EPP source code is freely available, results can be visualized via third party software and gates can be edited to adjust boundaries if necessary.

One of the advantages of EPP is its data-driven nature: Steps in the EPP algorithm are strictly based on data resolution; the order of parameter pairs considered in the gating strategy generated by EPP follows -for a given data set - the path where the best 2D resolution of phenotypically relevant subsets can be achieved. Although the algorithm's focus on the analysis of data resolution across parameters pairs in a 2-dimensional space may seem limiting at first, compared to the data mining power of high dimensionality reduction and clustering algorithms, this practical sequential 2D data analysis is still the procedure of choice by most cytometry analysts and the sole choice for most instrument-dedicated cytometry software. Sequential 2D analysis is used in cell sorters for the definition of sort gates, and it is also crucial to interpret and QC the phenotype of clusters generated via high dimensionality reduction and clustering algorithms. A preliminary thorough exploration of samples at the beginning of acquisition and using EPP, will also allow the adjustment of recording limits not to miss biological relevant subsets identified by the algorithm, enable robust statistics and clear definition of key cellular players in deep phenotyping and subset discovery projects.

Although readers lacking a background on programming or bioinformatics may find it difficult to understand the technical aspects of the EPP workflow and the metrics thoroughly described by the authors in the manuscript, they will be familiar with the approach taken by the algorithm to achieve a fast dissection of complex heterogeneous samples into unique subsets. As the authors mentioned in the last paragraph of the Discussion section (page 17), "EPP employs the same strategy for spatial reasoning as biologists do", only quicker and devoid of assumptions on the biology of samples, except for their intrinsic subset heterogeneity. Based on the comparison made by the authors between EPP's output and the manual analysis reported in OMIP 44, 47 and 77, the algorithm does a very good job in matching most of the reported reference phenotypes while identifying some that were excluded in the published studies. EPP could also highlight inconsistencies in phenotypes defined via manual approaches. EPP will be invaluable as a starting point for biological analysis and more: the possibility to integrate the algorithm's output into existing cytometry software will be particularly useful in cell sorting of very large polychromatic samples and rare samples, where speed of analysis leading to the isolation of viable, fully functional cells is as important to the success of the sort as the quality of the sample and the design of the polychromatic panel. By removing assumptions on the expected subsets contained in the sample, EPP will also strengthen the chances to discover new unsuspected cellular phenotypes. I congratulate the authors for sharing a methodology for the automatic, unbiased analysis of complex heterogeneous samples that will certainly improve the robustness and speed of cytometry data analysis.

That said, I have a few minor suggestions to improve the experience of the reader:

All the figures seem tiny, with tinnier fonts for axis and gate labels, barely visible if at all in the A4 printed document. Increasing the fonts and the overall size of the plots will allow the user to assess and confirm comments made in the document regarding EPP outcome vs. published OMIP results. The same applies to the tables.

We thank the reviewer for the very comprehensive and complimentary report.

(1) We revised the outputs for the figures and tables so that all but the tallest ones could be included with about the same width as the main text.

Figure legends and table descriptions are minimalist, providing a title but no further information enabling a clear understanding of the data presented, the meaning of columns or rows or the reason why some cells or fonts in the tables appear highlighted in green, blue, yellow, red or orange. Consider expanding the text to improve clarity or modifying the figures to remove unnecessary (distracting) highlights.

(2) The legends throughout have been amplified to make the relevant features understandable without reference to the main text so that it only needs to provide the interpretations.

The name of some of the gates in the EPP figures are confusing. On Figure 1, 2nd column first row (c2:r1), subsets are labeled as “CD3-CD14+” and “CD3++CD14”, instead of “CD3++CD14-“. In the same figure (c3:r2) eosinophils are labelled as “Eosinophils CD19+”. Isn’t CD19 a B cell pan marker? I understand that, from an algorithm perspective, a population appearing brighter along the CD19 detector will be classified as positive for the sake of just tagging, even when the intensity is entirely due to autofluorescence and not the result of fluorescence labelling. This could raise a few eyebrows and confuse the reader not aware of this. Would it be wise to add a comment along the text on the reason for the labelling and the potential impact of AF on fake phenotype classification?

(3) The EPP labels are constructed heuristically based on the FCS keywords and average signals in order to make them somewhat comprehensible and should not be taken as a biological assessment. “CD3++CD14” was an error and should be “CD3++CD14-“. In the case of “Eosinophils CD19+”, Eosinophils is drawn from the label given to the matched population in the reference publication and CD19+ indicates that the level in the CD19 dimension was used in gating relative to a statistically separable lower population although the difference could be just autofluorescence with no actual CD19 in either population. We have corrected the error in the figure and noted that the population labels indicate levels in the measurement dimensions for which biological interpretation may be required. [lines 203-206]

There’s no information on the fluorochrome used to label each marker in the EPP results. Is this a limitation of the algorithm or parameter labels can be modified at will at some point along the workflow?

(4) The dimension labels in the gating diagram figures were made from the \$PnN and \$PnS keywords in the FCS files. In OMIP-044 and OMIP-077 these keywords include the marker and primary detector name but not the fluorochrome name, so reagent details are found only in the paper itself and would have to be added by hand. We made a Fluorochrome-Dye-Detector list for the OMIP-044 and OMIP-077 reagents that is included as Supplementary Figure 20 to avoid renumbering the existing figures. Someone interested in extending their own biological interpretation of the data can find the fluorochrome labels there. [lines 223-224, 266-267 and Figure 20]

I couldn’t find information in the text of the time taken by EPP complete, which I take will depend on the size of the panel and the number of markers selected for the C++ based analysis. This information is important to appreciate the practicality of the procedure.

(5) We do discuss it briefly and another reviewer brings this up. The computations do not require any extraordinary hardware. Any computer suitable for running MATLAB, FlowJo or R routinely should be quite adequate. It is difficult to get accurate performance measurements on highly parallel processes running on general purpose computers. With the current development implementation, a significant amount of the time is taken in reading the ASCII encoded data files that should not be necessary if the method is called in process. We tested only with debug builds, not optimized release builds. For these reasons we did not do extensive timing analysis. In any case, as is EPP requires a few seconds clock time for the smallest of the sample experiments and a couple minutes for the mass cytometry one. Running the large, high dimension mass dataset with a qualifying threshold of zero (so that all pairs are examined) takes around 15 minutes. Performance should not be a barrier to deployment and use of EPP, and beyond that details on this didn’t seem useful. We have revised the text to make this clearer. [lines 169-172]

S.M. reports EPP takes 59 seconds to gate 27 dimensions of 500,000 events on a 6 core 2.9 GHz 7 year old MacBook.

Two final questions: given a defined cleaned data set, will EPP produce identical results across separate runs, or some variability could be expected?

(6) The current implementation is deterministic and always produces the same result. However, the underlying algorithm is not in the sense you mean, because it involves a pseudo random choice. A non-deterministic implementation made it hard to debug, so now we always start with the same random seed. However, if we did not do so or if using an independent implementation, the stochastic differences would be small on the scale of the tolerance of the polygon simplification we apply later. The results are not guaranteed to be numerically identical, but the differences will be within specified tolerance limits and will not be biologically significant (assuming a suitable tolerance and grid size were chosen for the application).

What would be the author's advice on the approach to be followed by researchers applying EPP to the simultaneous analysis to several samples differing in frequency and nature of subsets, including changes in marker expressions (naïve vs infected tissue samples)?

(7) This goes beyond what we have actually worked on, but it is indeed one of the big questions in moving forward from our current result. QFMatch[13] was designed to be a step in addressing this problem. Additional support structure needs to be developed to facilitate consistent analysis on multiple parallel samples.

2 Reply to the Reviewer #2

Moore et al. proposed a data-driven gating strategy, Exhaustive Projection Pursuit (EPP), to divide the samples on the 2-markers panel into two parts based on the density of measurement distribution. The split boundary is very flexible compared to the horizontal and vertical manual gating. The authors also made several comparisons between their EPP data-driven gating and the manual gating from several case studies. The auto-gating algorithm seems to function very well. However, several major components are missing for a publishable paper and a convincing tool. 1. A model diagram and reader-friendly description of the logic of modeling are missing to make the EPP concept and procedure clear. The concept that the gating boundaries went through the region where the sample density on the 2-marker panel is lowest is clear. However, the algorithm and model to achieve this goal are not clearly explained. Section 2.1 is challenging to follow and filled with jargon without clear definitions, such as “face,” “event,” “dip,” etc.

(1) We have included a schematic “EPP Algorithm” as Supplementary Figure 19 (to avoid renumbering the previous figures) and amended the text in several places to provide clearer definitions. [lines 36, 126-157]

2. The restriction of strictly splitting each 2-marker panel into two sections can limit the flexibility and generalization of using EPP. In the manual gating practice, it is common to use three or four gates to define three to four cell populations. For example, CD3+/-CD19+/- for major separation of B, T and monocytes. It seems to me that after splitting CD3 vs CD19 panel into two sections, EPP will not gate on CD3 and CD19 again. This will cause inconvenience for labs that have organized gating strategies for their tissue systems. Can authors relax this limitation of bi-partition? Or allow another round of further bi-partition on the same marker pairs?

(2) In fact, this choice is not limiting. Having used a pair of dimensions once does not preclude using it again. The data in the subpopulations are unchanged and the computation will discover, score and possibly select the same edges that were not selected previously. This computation is on some level redundant, but not worth the bookkeeping overhead to keep track of. This has always proven a tricky point to convey but we have tried to clarify the text. [lines 139-141]

In OMIP-044 EPP Gating-1 Supplementary Figure 9 Row 3 Columns 1 & 2 demonstrate exactly this case. Column 3 demonstrates why this is not always the right choice. A population is evident in Column 2 but EPP finds a better way to split it.

3. Authors compared EPP gating results with manual gating standards from each original publication of the case studies. However, it is not clear to me how the EPP gating sequence order is defined. It seems to me that EPP is a spatial splitting algorithm. However, in all of those EPP gating figures, such as Figure 1, how did EPP determine which pairs of markers to gate first and within which of the two splitting sections, further gating on which another pair of markers to split?

(3) The new figure “EPP Algorithm” (Supplementary Figure 19) should clarify this question. The order is determined by always taking the best separation found at each step. Unlike manual gating analyses where the order tends to reflect a person’s model for population relationships, the order of EPP gates is defined just by the current data and may be affected by small changes in the data that do not significantly alter the final populations.

A subsequent question is: did authors infer cell type and find the matching cell population with the manual gating results? For example, if a user is used to leverage CD3-/CD19+ to define B cell population. However, EPP split on CD3 vertically (CD3-, CD3+) and continues to gate CD3- population on other markers like CD14, CD16 etc, but not CD19 anymore. How should this user define B cells? In other words, if the EPP gating sequence order does not follow the user’s cell type annotation knowledge, how can the user find the matching cell populations split by EPP to define the cell type at the resolution they want correctly?

(4) This is an important question for interpretation of EPP results. EPP analysis is entirely data based and devoid of biological interpretation and content so that comparisons with biological expectations or functional assays are needed. For this manuscript we used QFMatch (Orlova[13]), which scores the proximity of populations in the measurement dimensions to align EPP populations with those identified in the referenced publications. The marker names for measurement dimensions are also drawn from those used in the reference

publications, and, as described above (Reviewer #1), the EPP +/- labels are generated heuristically. [lines 203-206]

4. Another major missing component is the comparison with existing auto-gating strategies. Upon a quick search, it seems that multiple auto-gating software has been launched, such as flowClust and OpenCyto. There is even a benchmarking paper to assess all kinds of automated flow cytometry data analysis techniques more than 10 years ago: <https://www.nature.com/articles/nmeth.2365>. Is anything novel and more powerful for EPP compared to all existing ones? I did not see any discussion in the current manuscript.

(5) We have added discussion of key features and advantages of EPP in relation to a number of other automated gating and clustering methods in the revised Discussion. [lines 455-484]

3 Reply to the Reviewer #3

Overall comment: Moore et al. developed EPP (Exhaustive Projection Pursuit), a tool to automate gating strategies for cytometry data. The authors have done a nice comparison of EPP across four cytometry datasets to validate its performance. They successfully identified populations that were missed in the initial OMIP paper. However, some sections can be more clearly described. It isn't immediately clear what the source of each gating strategy is, and some figures can be combined to improve the paper. Despite these shortcomings, the paper has some exciting results that would be beneficial to the greater cytometry community. Great job!

Major comments: •Create a figure that summarises how EPP works in section 2.1. There's a lot of information, but a figure would help clarify the workflow

We thank the reviewer for the positive assessment and for the set of valuable comments.

(1) We have added a workflow diagram "EPP Algorithm" as Supplementary Figure 19 (to avoid renumbering the previous figures) and revised the text in line with that diagram to clarify the EPP procedure [lines 126-157].

•It's not always clear which gating strategy has been generated by manual gating or by EPP. This needs to be made clearer throughout

(2) The figure legends and some points in the main text have been revised to clarify this. The manual gating figures are our recreations of the manual gating used in the reference publications since they did not provide the GatingML specifications that would allow exact reproduction of those gates.

Minor comments:

•At the end of section 2.1, the runtime of EPP is mentioned but nothing is detailed. Include a table (can be supplementary) that quantifies the run time based on number of cells. Also include information on hardware (particularly if a HPC is used)

(3) Reviewer #1 also asked about this. We have added some information in the text about requirements and run time [lines 169-172]. We explain why we do not include more details in the response to Comment (5) of Reviewer #1.

•Is Figure 1 generated using EPP or is it the gating strategy from the referenced paper? It's not immediately clear so re-word. It can also be combined with Figure 2

(4) Figure 1 shows the OMIP-077 EPP Gating Sequence. Our recreation of the reference gating is in Supplementary Figure 7. The figure legends include "EPP" for the EPP diagrams and "Recreation" for our recreations of the gating diagrams in the original publications.

•Section 2.2 needs a figure to better visualise what is being described. It may be appropriate to combine with section 2.1, and/or include the mathematics/functions behind the described calculations

(5) A diagram of a simple case has been included within Supplementary Figure 19. Section 2.2 is separate because we think the algorithm has not previously been published and needs to be documented, but the details are of interest to computer science and only the most general idea is needed to fully understand the biological implications for EPP itself.

•In section 2.3.1, "Our emulation of the reference gating scheme is shown in Figure 7". Do you mean this was the original gating strategy as proposed by the OMIP or is this the result of EPP? It should be re-worded for clarity

(6) Figure 7 is our best-estimated recreation of the gating scheme illustrated in Figure 1 of the OMIP-077 reference paper. We have changed the term "emulation" to "recreation" in the text and legends of the non-EPP gating figures.

•Figure 2: the EPP populations should be more clearly defined. Assuming Figure 1 contains the gating strategy developed by EPP, “7 Monocytes(CD16+)” are CD14++ whilst “Monocytes(CD16+CD19-)” are CD14-. These are intermediate and non-classical monocytes (respectively), whilst “6 Monocytes(CD16-)” (which are also CD14++) are classical monocytes. Better define the phenotypes of these populations

(7) The biological identifications of the EPP cell populations are in all cases drawn from the cell type assigned in the reference publication for the population best matched to the EPP population. Further naming of EPP phenotypes as they might be defined by biologists working in the field would require information not specified in the reference gating annotation. The CD and other markers used in the EPP population names are drawn from the reagent panels used in the reference publications. The +/- designations indicate EPP gating levels and are not necessarily definitive for the presence or absence of the markers. We have clarified this in the text [lines 203-206] and legends for figures 2, 3, 4, 5 and 18.

•Section 2.3.1 mentioned an opt-SNE plot in Figure 1 that doesn't exist

(8) The opt-SNE plot is only in Figure 1 of the OMIP-077 reference publication. Due to the stochastic aspects of the opt-SNE process, a new calculation for opt-SNE would not be expected to match the published one, so we did not attempt to recreate that portion of the original figure.

•Section 2.3.2: “The upper three rows in Figure 8 show our emulation of the main gating structure illustrated in Figure 1”. This suggests Figure 8 shows the gating strategy of EPP, but I don't think that's what is trying to be said? It's also not clear how this section relates to figure 1 (OMIP-077). This paragraph should be re-worded as it's not clear what the message is

(9) Updating from “emulation” to “recreation”, Figure 8 recreates the gating structure in Figure 1 of Mair, et al (i.e., the OMIP-044 publication). The figure legends will be clarified further, but the ones originally labeled “emulation” are our recreations of the published gating structures, and the ones including “EPP” show EPP gating.

•Section 2.3.3: “Our emulation of the reference gating in Figure 1 of OMIP-047”. Is “Figure 1” referring to the OMIP paper? If so, remove it (or call it something else) as it's confusing. Furthermore, if figures like Figure 11 are just copied from the OMIP paper they need to be correctly referenced/licensed

(10) “Our emulation” would be synonymous with “Our recreation”, and Figure 1 of OMIP-047 is in the OMIP paper. Since the reference gating figure is “Figure 1” in each of the four publications, we now specify explicitly which paper is involved whenever we refer to a Figure 1. None of our figures is an actual copy of a previously published figure.

•Mass cytometry figures 14-17: adjust the axes so you can more clearly see the cells backed up against the axes

(11) The scale was chosen to always yield the conventional top of scale value while placing small integer values in the lowest “decade” of the display. Since the mass data often includes a lot of true zero values and the low integer values are smeared in the data provided by the instrument, we have not identified a better way to visualize the low end of the distribution.

•T-SNE or UMAP plots would be useful for visualising the populations identified by EPP and how it compares to the reference OMIP dataset

(12) We have relied on the events matching procedure QFMatch for identifying the relationships among the EPP and reference cell populations. We have modified the text to make this clear. [lines 207-209]

•Make some of the figures supplementary

(13) Figures 7 to 18 are in a Supplementary Figures section where the header was on a different page from the first of the supplementary figures. We have revised the legends to state clearly which figures are supplementary.

•Is it recommended to run EPP on a concatenated file of all samples (which may be skewed by batch effect) or to run each sample separately (which may in turn produce different results)?

(14) We don't recommend this in general but that is what the referenced study did, so we did the same.

•Most packages being used to analyse cytometry data are in R or Python. Making EPP available in these other languages will increase the usage. It's great that it's implemented in FlowJo

(15) There is a MATLAB binding available now. We worked with FlowJo/BD and they were very helpful, but we were unable to make it work directly within FlowJo, due to limitations in the existing plugin framework. However, making it accessible is possible with further work. We had intended to produce an R package. Our build procedure was not compatible with the CRAN archive, but we think it is acceptable for BioConductor, but we have not had resources to carry that forward. The simple command line interface used by the MATLAB-FlowJo bridge is easily accessible from R and the output is JSON and CSV that existing R packages can read. Since it involves text files, it's not particularly efficient, but it is equivalent to what we used here.